# SAFLEX: SELF-ADAPTIVE AUGMENTATION VIA FEATURE LABEL EXTRAPOLATION

**Mucong Ding, Bang An, Yuancheng Xu, Anirudh Satheesh, Furong Huang**
Department of Computer Science
University of Maryland
`mcding@cs.umd.edu`

## ABSTRACT

Data augmentation, a cornerstone technique in deep learning, is crucial in enhancing model performance, especially with scarce labeled data. While traditional methods, such as hand-crafted augmentations, are effective but limited in scope, modern, adaptable techniques often come at the cost of computational complexity and are hard to fit into existing processes. In this work, we unveil an efficient approach that universally enhances existing data augmentation techniques by enabling their adaptation and refinement, thereby providing a significant and comprehensive improvement across all existing methods. We present **SAFLEX** (**S**elf-Adaptive **A**ugmentation via **F**eature **L**abel **EX**trapolation), an approach that utilizes an efficient bilevel optimization to learn the *sample weights* and *soft labels* of augmented samples. This is applicable to augmentations from any source, seamlessly integrating with existing upstream augmentation pipelines. Remarkably, SAFLEX effectively reduces the noise and label errors of the upstream augmentation pipeline with a marginal computational cost. As a versatile module, SAFLEX excels across diverse datasets, including natural, medical images, and tabular data, showcasing its prowess in few-shot learning and out-of-distribution generalization. SAFLEX seamlessly integrates with common augmentation strategies like RandAug and CutMix, as well as augmentations from large pre-trained generative models like stable diffusion. It is also compatible with contrastive learning frameworks, such as fine-tuning CLIP. Our findings highlight the potential to adapt existing augmentation pipelines for new data types and tasks, signaling a move towards more adaptable and resilient training frameworks.

## 1 INTRODUCTION

Data augmentation is a cornerstone in improving machine learning models, especially when labeled data is scarce. It enhances model performance by introducing varied training samples. Though traditional methods like rotation and cropping are widely used, they operate under a one-size-fits-all assumption that often falls short in the complexity of real-world data. The key is not just to augment data, but to do it in a way that does not mislead the learning process.

Recent work emphasizes the benefits of learned data augmentation, where techniques such as AutoAugment (Cubuk et al., 2019) and RandAugment (Cubuk et al., 2020) adapt to specific datasets and tasks. While promising, this area is still nascent and lacks a comprehensive framework to address diverse tasks and data nuances. Furthermore, selecting meaningful transformations remains a challenge, often relying on heuristics or domain expertise, which is especially problematic in specialized fields. Inappropriate transformations can harm model performance, underscoring the need for systematic selection. Amid the rise of image generation methods, such as diffusion models and other generative AI, an abundance of synthetic data is available but requires discerning use. A recent study, LP-A3 (Yang et al., 2022a), aims to generate "hard positive examples" for augmentation but risks introducing false positives that could mislead learning. Another recent work, Soft-Augmentation (Liu et al., 2023), introduces soft learning targets and loss reweighting to train on augmented samples but is primarily limited to improving image crop augmentation. The overarching need is for smarter, more adaptable data augmentation algorithms.

This paper proposes SAFLEX (Self-Adaptive Augmentation via Feature Label Extrapolation), which automatically learns the sample weights and soft labels of augmented samples provided by any given

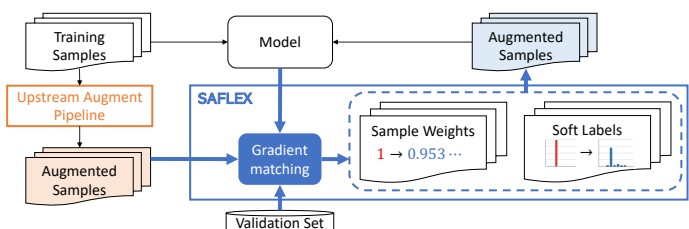

Figure 1: SAFLEX learns to adjust sample weights and soft labels of augmented samples from an upstream pipeline, aiming to maximize the model's performance on the validation set. While formulated as a bilevel optimization problem, it can be efficiently solved by linear programming with a gradient-matching objective. SAFLEX is a plug-in to the existing training framework.

upstream augmentation pipeline. Existing learnable augmentation methods that directly learn in the feature space (e.g., image space) often restrict augmentation scope due to differentiability needs and suffer from complicated training in high-dimensional spaces. Contrary to this, we advocate for learning only low-dimensional sample weights and soft labels for each augmented instance sourced from a pre-existing upstream augmentation pipeline like synthetic data generation. While upstream augmentation methods can sometimes alter labels or introduce noise, especially when creating samples outside the data distribution, our approach offers a mechanism to correct them. By calibrating sample weights and labels after augmentation, we considerably alleviate issues stemming from upstream augmentation methods. Without the complexity of learning augmentation transformations from scratch, this strategy ensures that augmentation is both diverse and consistent with the inherent data distribution, thereby fostering better generalization across various tasks. See Fig. 1 for a demonstration of our proposed SAFLEX.

We frame learning sample weights and soft labels as a bilevel optimization problem. This captures the interdependent nature of the model and its augmented data: the model's performance depends on the quality of the augmented data, which in turn is guided by the model itself (Bard, 2013). This new perspective advances our understanding of data augmentation, offering a theoretical framework that underpins its practical applications. Despite the bilevel nature of the problem, direct solutions are computationally infeasible for large-scale real-world applications. To mitigate this, we propose a streamlined, greedy, online, single-level approximation algorithm, which optimizes a gradient-matching objective to accelerate the learning process.

We conducted extensive empirical evaluations to highlight SAFLEX's superior performance and adaptability. On eight medical images (Yang et al., 2023), SAFLEX elevates popular augmentation techniques like RandAugment (Cubuk et al., 2020) and Mixup (Zhang et al., 2018), boosting performance by up to $3.6\%$. On seven tabular datasets, SAFLEX shows compatibility with categorical data and effectively enhances CutMix (Yun et al., 2019). Furthermore, SAFLEX improves image augmentations from diffusion models, yielding an average improvement of $1.9\%$ in fine-grained classification and out-of-distribution generalization against three diffusion-augmentation methods, harnessing on their pre-trained expertise. We also validate SAFLEX's integration with contrastive learning through a CLIP fine-tuning experiment. These findings underline its versatility across varied data types and learning tasks.

Our contributions are threefold:
(1) We unveil a novel parametrization for learnable augmentation complemented by an adept bilevel algorithm primed for online optimization.
(2) Our SAFLEX method is distinguished by its universal compatibility, allowing it to be effortlessly incorporated into a plethora of supervised learning processes and to collaborate seamlessly with an extensive array of upstream augmentation procedures.
(3) The potency of our approach is corroborated by empirical tests on a diverse spectrum of datasets and tasks, all underscoring SAFLEX's efficiency and versatility, boosting performance by $1.2\%$ on average over all experiments.

## 2 PROPOSED METHOD: SAFLEX

Our goal is to refine augmented samples from any upstream pipeline to enhance classifier generalization. The proposed methodology is founded on two pivotal questions: (1) Which aspects of the

augmented samples should be refined? (2) What approach should be taken to learn these refined samples? We start from these questions and defer the derivation of the algorithm to Section 3.

**Limitations of Augmentation Methods.** Data augmentation is pivotal in enhancing model generalization. However, its limitations, particularly the unintentional introduction of noise, can sometimes outweigh its benefits. For instance, consider the widespread use of random cropping on natural images. Although largely effective, there are times when this approach inadvertently omits task-relevant information, leading to unintended outcomes like false positives. This inherent noise creates a trade-off: under-augmentation may yield insufficient challenging examples, whereas over-augmentation can flood the dataset with misleading samples. As shown in Fig. 2a, reducing the noise in augmentation is the key to resolving the dilemma.

Noise in augmentation primarily arises from two fundamental challenges: (1) the deviation of augmented samples from the original data distribution and (2) the potential mislabeling of augmented samples. We shall envision augmentation as a method to harness prior knowledge in capturing the underlying data distribution. This distribution is encapsulated in the joint distribution, $\mathbb{P}_{XY}(x, y)$, where $x \in \mathcal{X}$ are features and $y \in \{1, \ldots, K\}$ represents labels, with $K$ indicating the number of classes. Breaking down this joint distribution: $\mathbb{P}_{XY}(x, y) = \mathbb{P}_X(x) \cdot \mathbb{P}_{Y|X}(y|x)$, we observe that the primary source of noise is associated with the feature distribution $\mathbb{P}_X(x)$, while the secondary source is tied to the conditional distribution $\mathbb{P}_{Y|X}(y|x)$. Addressing these challenges, our methodology is designed to integrate seamlessly with any upstream augmentation process, amending both types of errors post-augmentation, and considering the initial augmentation process as a separate, unchanged entity.

**Feature and Label Extrapolation.** A key concern in data augmentation pertains to addressing these two types of errors. Some prior works on learning augmentation (e.g., (Yang et al., 2022a)) attempted to reduce noise by fine-tuning augmented features, using them as initializations. Specifically, the aim was to derive a modified feature $x'$ that eliminates both error types. Yet, due to the high-dimensionality of feature space $\mathcal{X}$, manipulating $x$ is computationally burdensome.

A more efficient strategy is to handle the errors individually and abstain from modifying $x$. When encountering erroneous estimation of the feature distribution $\mathbb{P}_X(x)$, even if augmented samples lie in low-density areas, we can compensate by modulating the sample weights $w \in [0, 1]$ in the empirical risk minimization loss. Specifically, rarer augmented features are assigned decreased sample weights. For inaccuracies in estimating the conditional distribution $\mathbb{P}_{Y|X}(y|x)$, it's advantageous to modify the augmented label $y$ directly. We also propose transitioning from a hard class label to a soft one, denoted as $\mathbf{y}$, representing a probability mass across $K$ classes, residing in the $K$-dimensional simplex $\mathbf{y} \in \Delta^K$. The proposed refinement of augmented samples is depicted in Eq. (1). Remarkably, optimizing these sample weights and soft labels effectively mitigates errors resulting from varied augmentation methods across numerous classification challenges.

$$(x, y) \xrightarrow{\text{Upstream Augment}} (x^{\text{aug}}, y^{\text{aug}}) \xrightarrow{\text{SAFLEX}} \Big( \underbrace{w^{\text{aug}}}_{\text{sample weight} \in [0, 1]}, x^{\text{aug}}, \underbrace{\mathbf{y}^{\text{aug}}}_{\text{soft label} \in \Delta^K} \Big) \quad (1)$$

To elucidate, consider a hypothetical example in Fig. 2b. Envision a training sample from the green class (represented by a pronounced green dot). Upon applying a noise-prone augmentation, such as Gaussian perturbation in a 2D setting, the augmented sample could either (1) fall into a region with few validation samples regardless of their class, or (2) be overwhelmingly encompassed by validation samples from a different class. In the former case, it is judicious to reduce the sample weights since they might not be pivotal in discerning the conditional distribution. In the latter instance, the label of the augmented sample should be fine-tuned. This can entail a shift to a soft label to rectify or mitigate potential label inconsistencies, informed by patterns in the validation set.

**Bilevel Formulation.** The remaining question in our design is how to learn the sample weights and soft labels for augmented samples. The overarching goal of augmentation is enhancing model generalization. While the test set remains inaccessible, a prevalent approach is to fine-tune performance using a validation set. This methodology aligns with standard practices in hyperparameter optimization and is evidenced in learnable augmentation methods such as AutoAugment (Cubuk et al., 2019) and RandAugment (Cubuk et al., 2020). Given a neural network $f(\cdot) : \mathcal{X} \to \Delta^K$ (where we assume Softmax is already applied and the outputs are $\mathbb{L}_1$ normalized) with parameter $\theta$, let us denote the training set, validation set, and the set of augmented samples as $\mathcal{D}_{\text{train}}$, $\mathcal{D}_{\text{val}}$, and

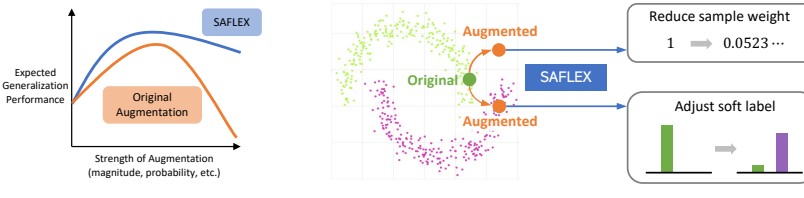

(a) Mitigating trade-offs.      (b) Addressing two types of errors.

Figure 2: **(a)** Under-augmentation can lead to a scarcity of hard positives, while over-augmentation can introduce an excess of false positives. Reducing the noise in augmentation helps resolve the dilemma. **(b)** Adjusting sample weights and recalibrating soft labels can address the two types of noises introduced by the augmentation process.

$\mathcal{D}_{\text{aug}}$, respectively. The ambition is to refine $\mathcal{D}_{\text{aug}}$ such that a model trained on the amalgamation of $\mathcal{D}_{\text{train}} \cup \mathcal{D}_{\text{aug}}$ optimizes performance on $\mathcal{D}_{\text{val}}$.

$$\min_{\mathcal{D}_{\text{aug}}, \theta} \mathcal{L}(\mathcal{D}_{\text{val}}, \theta) \quad \text{s.t.} \quad \theta \in \arg\min_{\theta'} \mathcal{L}(\mathcal{D}_{\text{train}} \cup \mathcal{D}_{\text{aug}}, \theta') \tag{2}$$

This scenario can be cast as a bilevel optimization problem as in Eq. (2), where $\mathcal{D}_{\text{aug}}$, the set of augmented samples with parametrized by sample weights and soft labels, and the model parameters $\theta$ are learnable. The conventional model training constitutes the inner level, while the quest to identify optimal augmented samples $\mathcal{D}_{\text{aug}}$, which minimize the validation loss post-inner level training, establishes the outer problem. Such a paradigm inherently transforms learnable augmentation into bilevel optimization. Intriguingly, much of the existing literature on learnable augmentation eschews this representation. The primary reservations stem from concerns related to efficiency and differentiability. Notably, works such as (Mounsaveng et al., 2021; 2023) are among the sparse few to apply bilevel optimization for augmentation learning, yet their focus remains constricted to affine transformations. In contrast, our approach sidesteps the modification and modeling of feature augmentation, obviating the challenge of differentiability. The low-dimensional nature of sample weights and soft labels potentially simplifies the learning process. In subsequent sections, we demonstrate that, under benign approximations, we can adeptly navigate the bilevel problem, determining the apt sample weights and soft labels within a singular step for each training iteration.

## 3    Algorithm

We now develop an algorithm for the bilevel problem described in Eq. (2).

**The Greedy Approach.** Bilevel optimization is notoriously challenging, often necessitating nested loops, which introduces significant computational overhead. Upon inspecting Eq. (2), it becomes evident that an essential characteristic of the problem — the training dynamics of the model — has been understated. In standard practice, augmented samples are typically generated during model training for each minibatch across all iterations. Therefore, the actual problem deviates from Eq. (2) in two ways: (1) different batches of augmentation may influence the learned parameters differently, and the model is not trained on a cumulative set of augmented samples, and conversely, (2) the learned parameters are affected differently by the refined augmented samples across batches, implying that augmentation should be optimized with respect to the corresponding model parameters.

To incorporate model optimization dynamics, we should reformulate the problem on a finer scale: Given the model parameter $\theta_{t-1}$ at an intermediate training step, how can we determine the batch of refined augmented samples, $\mathcal{D}_{\text{aug}}^{\text{batch}}$? Through a greedy approach, we posit that the granular objective is to minimize the validation loss after a single update, denoted as $\mathcal{L}(\mathcal{D}_{\text{val}}, \theta_t)$, where $\theta_t$ is the model parameter updated from $\theta_{t-1}$.

$$\min_{\mathcal{D}_{\text{aug}}^{\text{batch}}, \theta_t} \mathcal{L}(\mathcal{D}_{\text{val}}, \theta_t) \quad \text{s.t.} \quad \theta_t = \theta_{t-1} - \alpha \cdot \nabla_\theta \mathcal{L}(\mathcal{D}_{\text{train}}^{\text{batch}} \cup \mathcal{D}_{\text{aug}}^{\text{batch}}, \theta_{t-1}) \tag{3}$$

This micro-perspective of Eq. (2) is represented in Eq. (3), where the batch of augmented samples $\mathcal{D}_{\text{aug}}^{\text{batch}} = \{(w_1^{\text{aug}}, x_1^{\text{aug}}, \mathbf{y}_1^{\text{aug}}), \dots, (w_B^{\text{aug}}, x_B^{\text{aug}}, \mathbf{y}_B^{\text{aug}})\}$ is parametrized by the set of sample weights $(w_1^{\text{aug}}, \dots, w_B^{\text{aug}})$ and soft labels $(\mathbf{y}_1^{\text{aug}}, \dots, \mathbf{y}_B^{\text{aug}})$.

As a direct consequence, if the inner loop uses a first-order optimizer like SGD (as assumed), this significantly eases the optimization task. The emergent problem is no longer bilevel. With the analytical solution of the "inner problem" at our disposal, we can integrate the formula for $\theta_t$ into the outer objective, $\mathcal{L}(\mathcal{D}_{\text{val}}, \theta_t)$, converting it into a single-level problem.

**Efficient Solution.** We next derive an algorithm for efficiently addressing Eq. (3). Crucially, due to the linearity of the loss function $\mathcal{L}(\cdot, \theta_{t-1})$ with respect to datasets and the inherent linearity of gradient computation, the gradient vector for the combined training and augmentation batch linearly relates to the sample weights and soft labels, assuming the sample-wise loss function, such as the cross-entropy loss, behaves linearly with respect to sample weights and soft labels. Validating this, the cross-entropy loss is indeed linear concerning these variables, a typical characteristic based on their definitions.

By approximating the validation loss $\mathcal{L}(\mathcal{D}_{\mathrm{val}}, \theta_t)$ up to the first order around parameter $\theta_t$, we can recast Eq. (3) as a linear programming problem. The objective seeks to maximize the inner product of the gradient vectors on the combined train and augmented batch and the validation batch, effectively yielding a gradient-matching loss (Zhao et al., 2020). Here, both the objective and normalization constraints linearly correspond to our learnable variables: sample weights and soft labels. The derivation is provided in Appendix A, and we summarize the solution in the subsequent notations and theorem.

**Notation 1.** *Let the Jacobian matrix of logits with respect to the model parameter be* $\nabla_\theta f(x^{\mathrm{aug}})\mid_{\theta=\theta_{t-1}} \in \mathbb{R}^{K \times m}$ *and the gradient vector on the validation set be* $\nabla_\theta \mathcal{L}(\mathcal{D}_{\mathrm{val}}, \theta_{t-1}) \in \mathbb{R}^m$, *where $m$ is the parameter count and $K$ is the class count. The Jacobian-vector product is denoted as* $\mathbf{\Pi} = \nabla_\theta f(x^{\mathrm{aug}})\mid_{\theta=\theta_{t-1}} \nabla_\theta \mathcal{L}(\mathcal{D}_{\mathrm{val}}, \theta_{t-1}) \in \mathbb{R}^K$, *which can be computed efficiently.*

**Theorem 1** (Solution of Eq. (3)). *The approximated soft label solution is* $\mathbf{y} = OneHot\left(\arg\max_k [\mathbf{\Pi}]_k\right)$, *where $OneHot(\cdot)$ denotes one-hot encoding, and the sample weight solution is $w = 1$ if $\sum_{k=1}^K [\mathbf{\Pi}]_k \geq 0$; otherwise, $w = 0$.*

Theorem 1 illustrates that an effective approximation of Eq. (3) is computationally efficient. The gradient inner product, $\mathbf{\Pi}$, a Jacobian-vector product, is readily computed alongside standard back-propagation on the combined training and augmentation batch. While determining the validation set gradient vector mandates an additional back-propagation step, we can approximate the gradient vector for the complete validation set, $\nabla_\theta \mathcal{L}(\mathcal{D}_{\mathrm{val}}, \theta_{t-1})$, using a minibatch gradient, $\nabla_\theta \mathcal{L}(\mathcal{D}_{\mathrm{val}}^{\mathrm{batch}}, \theta_{t-1})$. Despite necessitating a solution for Eq. (3) at every iteration, our efficient SAFLEX algorithm incurs minimal computational overhead.

A notable takeaway from Theorem 1 is that while we aim to learn continuous sample weights (in $[0,1]$) and soft labels (in $\Delta^K$), the derived solutions consistently yield discrete values: either 0 or 1 and one-hot vectors. This consistency does not signify a coarse approximation, especially considering we resolve Eq. (3) with a $O(\alpha)$ tolerance, where $\alpha$ is typically small. Nonetheless, this characteristic could potentially impact model generalization in under-parameterized scenarios.

**Generalization Aspects.** Let's interpret and examine the solution provided by Theorem 1 from a generalization standpoint, which is our primary objective. The loss function's linearity helps understand Theorem 1.. Given that Eq. (3) is a linear program with straightforward normalization constraints, we effectively form a linear combination of $KB$ gradient vectors (each pertaining to a logit of the augmented sample), with $B$ representing the augmented batch size, to approximate the $m$-dimensional validation gradient vector. The total constraints sum up to $B + 1$. If these $KB$ gradient vectors are linearly independent, we can always align the combined gradient vector with the validation gradient vector when the degree of freedom, $B + KB - (B+1)$, is greater than or equal to the gradient vector dimension, $m$. This is represented by the condition $KB > m$. Such a scenario, exceedingly under-parametrized, is rare in deep learning. If the combined gradient vector consistently aligns with the validation gradient vector, training with SAFLEX will approximate training on the combined training and validation sets, potentially limiting the generalization improvements.

To enhance generalization, it is essential to circumvent the challenges of the under-parametrized paradigm, even if we are not closely approaching it. Here, we suggest two modifications to the solution given by Theorem 1:

**1. Encouraging Retention of the Original Label.** We can introduce a minor constant penalty term to the gradient inner product to incentivize retaining the augmented sample's original label. Thus, we substitute $\mathbf{\Pi}$ with $\mathbf{\Pi} + \beta \mathbf{e} y^{\mathrm{aug}}$, where $\mathbf{e} y^{\mathrm{aug}}$ is a one-hot vector with a value of 1 at the $y^{\mathrm{aug}}$-th position. If no other entry in $\mathbf{\Pi}$ exceeds $[\mathbf{\Pi}]_{y^{\mathrm{aug}}}$ by a margin of at least $\beta$, the learned label remains unaltered. This approach proves especially valuable when the validation set is of limited size.

**2. Substituting** $\arg\max$ **with Gumbel-SoftMax.** Our current solution invariably yields hard labels. This can sometimes manifest as an excessive degree of confidence, particularly when $\mathbf{\Pi}$ contains

multiple significant entries. To alleviate this, we can employ the Gumbel-SoftMax function to introduce a "softening" effect to the learned labels, adding a measure of stochasticity. Hence, we have $\mathbf{y} = \text{softmax}\big((\mathbf{\Pi} + \beta\mathbf{e}_{y^{\text{aug}}} + \mathbf{g})/\tau\big)$, where $\mathbf{g}$ consists of i.i.d. random variables sourced from $\text{Gumbel}(0, 1)$. Typically, unless specified otherwise, we opt for a relatively low fixed temperature value, $\tau = 0.01$.

**The pseudo-code of SAFLEX** for cross-entropy loss is shown as Algorithm 1.

---

**Algorithm 1:** SAFLEX (Cross-Entropy Loss, Single batch).

---

**Input:** Neural network $f(\cdot) : \mathcal{X} \to \Delta^K$ (softmax applied on outputs) with parameters $\theta$, upstream augmented batch $\{(x_1^{\text{aug}}, y_1^{\text{aug}}), \ldots, (x_B^{\text{aug}}, y_B^{\text{aug}})\}$, validation batch $\mathcal{D}_{\text{val}}^{\text{batch}} = \{(x_1^{\text{val}}, y_1^{\text{val}}), \ldots, (x_{B'}^{\text{val}}, y_{B'}^{\text{val}})\}$, penalty coefficient $\beta$, temperature $\tau$.

1  Compute the gradient vector for the validation batch $\nabla_\theta \mathcal{L}(\mathcal{D}_{\text{val}}^{\text{batch}}, \theta)$.

2  **for** $i = 1, \ldots, B$ **do**                    // The actual implementation is vectorized.

3  |    Determine the gradient inner product $\mathbf{\Pi}_i = \nabla_\theta f(x_i^{\text{aug}}) \nabla_\theta \mathcal{L}(\mathcal{D}_{\text{val}}^{\text{batch}}, \theta)$ via Jacobian-vector product.

4  |    Apply Gumbel-SoftMax to get $\mathbf{y}_i = \text{softmax}\big((\mathbf{\Pi}_i + \beta\mathbf{e}_{y_i^{\text{aug}}} + \mathbf{g})/\tau\big)$, where $e_{y_i^{\text{aug}}} \in \mathbb{R}^K$ is one-hot at $y_i^{\text{aug}}$, and $\mathbf{g}$ consists of i.i.d. random variables taken from $\text{Gumbel}(0, 1)$.

5  |    Set $w_i = 1$ if $\mathbf{\Pi}_i \cdot \mathbf{y}_i \geq 0$, otherwise set $w_i = 0$.

6  Renormalize the sample weights $w_1, \ldots, w_B$ to sum to 1.

7  **return** *Sample weights* $w_1^{\text{aug}}, \ldots, w_B^{\text{aug}}$, *and soft labels* $\mathbf{y}_1^{\text{aug}}, \ldots, \mathbf{y}_B^{\text{aug}}$.

---

**SAFLEX for Contrastive Learning.** We conclude this section by discussing to encompass the generalization of the proposed method for certain contrastive learning losses, as illustrated in Eq. (12) and Eq. (13). Notably, the latter is utilized for CLIP training. In the realm of contrastive learning, labels are not conventionally defined. Yet, one can perceive the contrastive training objectives in Eq. (12) and Eq. (13) as proxy classification tasks. Here, we posit that the batch of size $B$ can be construed as containing $B$ classes: one positive example coupled with $B-1$ negative examples. This interpretation paves the way to introduce the notion of (soft) labels over this surrogate classification task with its $B$ distinct classes.

Under this paradigm, the loss function remains linear concerning the soft labels and sample weights, making the methodology in Theorem 1 applicable. The sole requisite modification pertains to the gradient inner product's definition. Rather than employing gradients from the cross-entropy logits, $\nabla_\theta f(x^{\text{aug}})$, we substitute them with gradients corresponding to the contrastive learning logits.

## 4  RELATED WORKS

Traditional data augmentation techniques such as random flipping and cropping (Krizhevsky et al., 2017; Simard et al., 2003; Shorten & Khoshgoftaar, 2019) are hand-crafted and static, unlike our adaptive SAFLEX method that tunes sample weights based on validation performance. Autonomous approaches like AutoAugment (Cubuk et al., 2019; Lim et al., 2019; Ho et al., 2019; Mounsaveng et al., 2021; 2023) learn transformations but are restricted in scope, primarily focusing on affine transformations. Generative methods employing GANs or diffusion models (Odena et al., 2017; Sankaranarayanan et al., 2018; Huang et al., 2018; He et al., 2022; Shipard et al., 2023; Dunlap et al., 2023; Trabucco et al., 2023) can inadvertently alter class-relevant features, which our method avoids by adaptively adjusting sample weights. Research on adversarial perturbations (Goodfellow et al., 2015; Yang et al., 2022a;b; Ho & Nvasconcelos, 2020) and noise-robust learning (Han et al., 2018; Lang et al., 2022; Thulasidasan et al., 2019; Konstantinov & Lampert, 2019; Gao et al., 2022; Ma et al., 2018; Kremer et al., 2018) address similar problems but often suffer from complexity and stability issues, which we mitigate by our principled approach to weight adjustment. Recently, Soft-Augmentation (Liu et al., 2023) also proposes to use soft labels and sample weights to train on augmented samples. However, it implements a specific formula to generate them based on the strength parameter of upstream augmentations. This limits the applicability of Soft-Augmentation mostly to crop augmentation on images. Wang et al. (2023) introduce self-adaptive augmentation within the meta-learning framework, MetaMix, which improves the corruption robustness of continual learning models. Bhattarai et al. (2020) propose a progressive sampling strategy for GAN synthetic data, while Caramalau et al. (2021) introduce a sequential graph convolutional network for active learning. Our work extends these findings by developing a novel sampling and purifying method for augmented data that is specifically designed to improve the performance of downstream tasks.

For a more detailed discussion of related works, please refer to Appendix B.

## 5 EXPERIMENTS

We validate the effectiveness of SAFLEX under four very different learning scenarios: (1) adapting augmentations to medical images, (2) refining augmentations for tabular data, (3) purifying diffusion-model-generated augments, and (4) applying to contrastive fine-tuning. Experimental setups and implementation details are provided in Appendix C.

**Adapting Augmentations to Medical Images.** Unlike natural images, medical images often carry quantitative information (e.g., encoded as color) and objects without a canonical orientation. While we usually lack the domain knowledge to design effective heuristic augmentation transformations for these images, applying augmentation pipelines designed for natural images, such as RandAugment (Cubuk et al., 2020), can sometimes degrade performance in the medical context (Yang et al., 2022a). Consequently, we investigate whether SAFLEX can adapt these augmentation pipelines for medical images.

We assess multi-class classification across eight medical image datasets from MedMNIST (Yang et al., 2023), with each dataset comprising 10K to 236K 28×28 images and 4 to 11 classes. In line with (Yang et al., 2021), we train a ResNet-18 model (He et al., 2016) using the Adam optimizer (Kingma & Ba, 2014) for 100 epochs. For upstream augmentation, we utilize the widely-adopted RandAugment (Cubuk et al., 2020) and Mixup (Zhang et al., 2018) methods. Test accuracies are presented in Table 1, highlighting that SAFLEX significantly enhances the performance of both RandAugment and Mixup. It's noteworthy that SAFLEX, when combined with basic upstream augmentations as shown in Table 1, achieves better performance than Soft-Augmentation (Liu et al., 2023), and comparable or superior performance than the adversarial-perturbation-based augmentation, LP-A3 (Yang et al., 2022a). The latter not only takes significantly longer to train but also demands careful hyperparameter tuning. For a comprehensive view, Soft-Augmentation's performance and LP-A3's performance on the MedMNIST datasets can be found in Appendix C.

| Method | Path | Derma | Tissue | Blood | OCT | OrganA | OrganC | OrganS |
|---|---|---|---|---|---|---|---|---|
| No Aug | $94.34 \pm 0.18$ | $76.14 \pm 0.09$ | $68.28 \pm 0.17$ | $96.81 \pm 0.19$ | $78.67 \pm 0.26$ | $94.21 \pm 0.09$ | $91.81 \pm 0.12$ | $81.57 \pm 0.07$ |
| RandAug | $93.52 \pm 0.09$ | $73.71 \pm 0.33$ | $62.03 \pm 0.14$ | $95.00 \pm 0.21$ | $76.00 \pm 0.24$ | $94.18 \pm 0.20$ | $91.38 \pm 0.14$ | $80.52 \pm 0.32$ |
| SAFLEX (w/ RandAug) | $\mathbf{95.11 \pm 0.14}$ | $76.69 \pm 0.33$ | $64.32 \pm 0.18$ | $96.91 \pm 0.15$ | $\mathbf{79.63 \pm 0.28}$ | $\mathbf{95.32 \pm 0.29}$ | $92.10 \pm 0.21$ | $\mathbf{82.85 \pm 0.42}$ |
| Mixup | $92.98 \pm 0.19$ | $75.22 \pm 0.45$ | $66.62 \pm 0.31$ | $96.28 \pm 0.23$ | $77.93 \pm 0.41$ | $94.12 \pm 0.35$ | $90.76 \pm 0.28$ | $80.99 \pm 0.21$ |
| SAFLEX (w/ Mixup) | $93.71 \pm 0.37$ | $\mathbf{76.94 \pm 0.51}$ | $\mathbf{68.31 \pm 0.43}$ | $\mathbf{97.21 \pm 0.35}$ | $79.54 \pm 0.44$ | $95.06 \pm 0.31$ | $\mathbf{92.73 \pm 0.53}$ | $82.14 \pm 0.27$ |

Table 1: On **medical images**, SAFLEX significantly enhances the performance of RandAugment and Mixup across eight medical image datasets from MedMNIST.

In terms of efficiency, SAFLEX, designed as an augmentation plug-in, requires only a single-step update per iteration. It only extends the average wall-clock time of a training epoch by roughly 42% in this experiment; see Appendix C for details.

**Refining Augmentations for Tabular Data.** Tabular data typically encompasses heterogeneous features that include a blend of continuous, categorical, and ordinal values. The presence of discrete features constrains the space of potential transformations. Furthermore, the domain knowledge to design invariant, label-preserving transformations is often absent. One of the few traditional augmentation techniques directly applicable to tabular data is CutMix (Yun et al., 2019), which substitutes a portion of continuous or discrete features with values from other randomly chosen samples (see Appendix C for implementation details). However, studies suggest that CutMix, with a relatively small augmentation probability like 0.1, struggles to bolster tabular classification performance (Onishi & Meguro, 2023). Conversely, a higher augmentation probability can introduce excessive noise, potentially downgrading the performance. This leads us to explore whether SAFLEX can mitigate the noise from CutMix and enhance classification performance.

Our experiments span seven tabular datasets varying in size (from 452 to 494K) and feature types (from exclusively continuous features to predominantly discrete ones); detailed dataset information and statistics are available in Appendix C. Except for the *Volkert* dataset, which involves 10-way classification, all other datasets focus on binary classification. Notably, some datasets, like *Credit*,

exhibit a significantly skewed class distribution (e.g., only 0.17% positive). We consider backbone models such as the sample Multilayer Perceptron (MLP) with two hidden layers and 256 neurons each and tranformer-based models like SAINT (Somepalli et al., 2022) (without contrastive pre-training). These models undergo training with dropout (Srivastava et al., 2014) and, in certain cases, batch normalization, for 200 epochs.

| Method | Model | Appetency | Arrhythmia | Click | Credit | QASR | Shrutime | Volkert |
|---|---|---|---|---|---|---|---|---|
| No Aug | MLP | 49.03 ± 0.01 | 81.53 ± 0.03 | 52.54 ± 0.04 | 66.91 ± 0.03 | 91.84 ± 0.02 | 86.27 ± 0.04 | 61.14 ± 0.05 |
| CutMix | MLP | 48.98 ± 0.03 | 81.57 ± 0.05 | 52.59 ± 0.09 | 73.68 ± 0.08 | 91.87 ± 0.02 | 86.39 ± 0.05 | 61.20 ± 0.02 |
| SAFLEX (w/ CutMix) | MLP | **51.04 ± 0.09** | **83.02 ± 0.06** | **52.81 ± 0.06** | **74.61 ± 0.15** | **92.69 ± 0.13** | **86.90 ± 0.10** | **61.51 ± 0.05** |
| No Aug | SAINT | 78.90 ± 0.03 | 83.90 ± 0.01 | 65.72 ± 0.04 | 79.49 ± 0.05 | 98.18 ± 0.04 | 87.53 ± 0.04 | 66.82 ± 0.05 |
| CutMix | SAINT | 81.05 ± 0.07 | **85.32 ± 0.09** | 65.77 ± 0.04 | 79.71 ± 0.08 | **98.61 ± 0.06** | 87.61 ± 0.07 | 68.23 ± 0.10 |
| SAFLEX (w/ CutMix) | SAINT | **81.33 ± 0.14** | 85.27 ± 0.14 | **66.12 ± 0.09** | **79.93 ± 0.17** | 98.59 ± 0.21 | **87.93 ± 0.13** | **68.91 ± 0.17** |

Table 2: On **tabular data**, SAFLEX outperforms the upstream augmentation method, CutMix, across diverse tabular datasets using either MLP or SAINT as the backbone models.

Table 2 shows that SAFLEX almost consistently enhances the performance of CutMix across all datasets, regardless of whether the MLP or SAINT model is used. This improvement is especially noticeable with the MLP backbone, which is typically more intricate to train, and on datasets abundant in discrete features, such as *Click* and *Shrutime*, where CutMix tends to inject more noise. Notably, the *Volkert* dataset demonstrates a considerable performance impovement, potentially attributed to the fact that it has 10 classes where soft labels might be more useful.

**Purifying Diffusion-Model-Generated Augments.** Recent research (Dunlap et al., 2023; Trabucco et al., 2023) has advocated the application of diffusion models for image editing via text prompts. Compared to traditional augmentation techniques, images produced by pretrained diffusion models maintain task-specific details while offering enhanced domain diversity, as dictated by the prompts. Diffusion-model-generated augmentations have been found particularly efficacious in fine-grained classification and out-of-distribution (OOD) generalization tasks (Dunlap et al., 2023). However, these diffusion models occasionally generate subtle image alterations, potentially corrupting class-essential information, thus underscoring the necessity for noise reduction (Dunlap et al., 2023). In this context, we probe the capability of SAFLEX to enhance the purity of diffusion-model-generated augmentations, aiming for improved classification outcomes.

In our experimentation, we adhere to the setups in (Dunlap et al., 2023). We assess SAFLEX using diffusion-model-generated images derived from two distinct approaches: (1) The *Img2Img* approach involves an image encoder that first converts a given image into a latent representation. Subsequently, employing a diffusion model (specifically, Stable Diffusion v1.5 (Rombach et al., 2022) for this experiment), this latent representation undergoes a series of prompt-conditioned transformations. Ultimately, the altered representation is decoded, yielding an augmented image reflecting the modifications stipulated in the prompt. Notably, the diffusion model may or may not undergo fine-tuning (*w/ and w/o finetune*) on the dataset in question. (2) The *InstructPix2Pix* approach (Brooks et al., 2023) accepts an image and an edit instruction sampled (e.g., "position the animals within the forest") and outputs a correspondingly modified image. InstructPix2Pix is a conditional diffusion model pretrained on a dataset containing paired images and their associated edit instructions.

Our evaluation encompasses two tasks: (1) Fine-grained classification on a CUB dataset subset (Wah et al., 2011) (featuring 25 images per category). (2) OOD generalization on an iWildCam subset from the Wilds dataset (Koh et al., 2021) (consisting of over 6,000 images and simplified to 7-way classification). We use a ResNet-50 model (He et al., 2016) pretrained on ImageNet (Deng et al., 2009). For comparison, we also consider data generated solely from text (*Text2Img*) and the *RandAugment* method as baselines.

Results, as depicted in Table 3, affirm that SAFLEX consistently elevates the performance of all three diffusion-model-generated augmentation techniques, across both fine-grained classification and OOD generalization tasks. Notably, the performance boost is more prominent within the OOD generalization task, where feature and label distortions are particularly detrimental. We confirm that SAFLEX is useful to refine diffusion-model-generated augmentations, leading to enhanced classification accuracy.

| Task | No Aug | RandAug | Text2Img | InstructPix2Pix | | Img2Img (w/o finetune) | | Img2Img (w/ finetune) | |
|------|--------|---------|----------|-----------------|---|------------------------|---|----------------------|---|
| | — | — | — | w/o SAFLEX | w/ SAFLEX | w/o SAFLEX | w/ SAFLEX | w/o SAFLEX | w/ SAFLEX |
| Fine-Grained Classification | $68.60 \pm 0.16$ | $71.26 \pm 0.52$ | $69.68 \pm 0.97$ | $71.38 \pm 0.91$ | $\mathbf{72.34 \pm 0.59}$ | $71.25 \pm 0.86$ | $\mathbf{73.22 \pm 0.63}$ | $72.01 \pm 1.24$ | $\mathbf{73.61 \pm 0.78}$ |
| OOD Generalization | $57.19 \pm 1.13$ | $61.34 \pm 2.72$ | $64.53 \pm 3.01$ | $67.29 \pm 1.96$ | $\mathbf{69.92 \pm 0.88}$ | $70.65 \pm 1.50$ | $\mathbf{72.61 \pm 1.44}$ | $70.49 \pm 1.21$ | $\mathbf{72.83 \pm 0.92}$ |

Table 3: For **diffusion-model-generated augmentations**, SAFLEX enhances the fine-grained classification and OOD generalization performance for various diffusion-generation methods.

**Applying to Contrastive Fine-Tuning.** We next shift our focus from the empirical risk minimization (ERM) framework utilizing cross-entropy loss, as demonstrated in the prior scenarios. To test the adaptability and compatibility of SAFLEX with contrastive loss, we turn to a contrastive fine-tuning paradigm termed "Finetune Like You Pretrain" (FLYP)(Goyal et al., 2023). This methodology offers a straightforward yet potent means to fine-tune pretrained image-text models, including notable ones like CLIP (Radford et al., 2021). Remarkably, by simply fine-tuning classifiers through the initial pretraining contrastive loss (refer to Eq. (13)), FLYP achieves uniformly better classification performance. This entails constructing prompts from class labels and subsequently minimizing the contrastive loss between these prompts and the image embeddings within the fine-tuning set.

Our experimentation adopts the framework presented in (Goyal et al., 2023). Specifically, we fine-tune a CLIP model equipped with a ViT-B/16 encoder on the full iWildCam dataset from Wilds (Koh et al., 2021). Post fine-tuning, we adopt a strategy from (Goyal et al., 2023) that linearly interpolates weights between the pretrained and the fine-tuned checkpoints to optimize in-distribution (ID) performance. As our upstream augmentation technique, we employ RandAugment, following hyperparameter setups as described in (Koh et al., 2021). For handling the CLIP contrastive loss, we apply our tailored algorithm, detailed in Section 3. For an in-depth understanding, please refer to Appendix A and Appendix C.

| Task | | Zero-Shot | LP-FT | FLYP | FLYP+RandAug | FLYP+SAFLEX (w/ RandAug) |
|------|---|-----------|-------|------|--------------|--------------------------|
| ID | w/o Ensembling | $8.7 \pm 0.0$ | $49.7 \pm 0.5$ | $52.2 \pm 0.6$ | $52.4 \pm 0.8$ | $\mathbf{52.7 \pm 0.7}$ |
| OOD | | $11.0 \pm 0.0$ | $34.7 \pm 0.4$ | $35.6 \pm 1.2$ | $36.3 \pm 1.4$ | $\mathbf{36.9 \pm 1.5}$ |
| ID | w/ Ensembling | $8.7 \pm 0.0$ | $50.2 \pm 0.5$ | $52.5 \pm 0.6$ | $52.6 \pm 1.0$ | $\mathbf{53.0 \pm 0.7}$ |
| OOD | | $11.0 \pm 0.0$ | $35.7 \pm 0.4$ | $37.1 \pm 1.2$ | $37.6 \pm 0.9$ | $\mathbf{37.8 \pm 1.1}$ |

Table 4: Applied to **contrastive fine-tuning of CLIP using FLYP** (Goyal et al., 2023), SAFLEX also enhances the performance of standard image augmentations like RandAugment.

As evidenced in Table 4, incorporating RandAugment alongside FLYP yields favorable outcomes. Moreover, the introduction of SAFLEX amplifies performance gains for both ID and OOD tasks, irrespective of whether ensembling is applied. This observation is particularly noteworthy, as it demonstrates that SAFLEX is compatible with contrastive loss, which is a key component of many training paradigms, including self-supervised learning.

## 6 CONCLUSIONS

Our study presents SAFLEX, a novel solution to current challenges in data augmentation. At its core, SAFLEX offers a paradigm shift from traditional, one-size-fits-all augmentation strategies to a more adaptive, data-driven approach. It allows for the learning of low-dimensional sample weights and soft labels for each augmented instance, thereby circumventing the complexities and limitations inherent in direct augmentation feature learning. Our method demonstrates universal compatibility, underscoring its vast potential for diverse data types in learning scenarios. Extensive empirical evaluations confirm SAFLEX's prowess, proving its adaptability from medical imaging contexts to the nuances of tabular and natural image datasets. While SAFLEX demonstrates promising results, there are certain factors to consider for optimal performance. A substantial and high-quality validation set is beneficial. A suboptimal set could limit its effectiveness. Additionally, the type of upstream augmentation methods selected plays a role, as it impacts the overall performance of SAFLEX. Our approach also entails some computational overhead due to frequent gradient evaluations. These considerations will be the focus of future studies to further refine the methodology. In essence, SAFLEX stands as a testament to the advancements in learnable data augmentation, ushering in a more adaptive and customized era of data-centric AI.

ACKNOWLEDGMENTS

Ding, An, Xu, Satheesh, and Huang are supported by National Science Foundation NSF-IIS-2147276 FAI, DOD-ONR-Office of Naval Research under award number N00014-22-1-2335, DOD-AFOSR-Air Force Office of Scientific Research under award number FA9550-23-1-0048, DOD-DARPA-Defense Advanced Research Projects Agency Guaranteeing AI Robustness against Deception (GARD) HR00112020007, Adobe, Capital One and JP Morgan faculty fellowships.

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
