# A  METHOD AND ALGORITHM DETAILS

In this appendix, we provide more details about the proposed method and algorithm. We first show the derivation details behind results in Theorem 1. Then, we provide more details about the SAFLEX algorithm on contrastive losses discussed at the end of Section 3.

*Proof of Theorem 1:* As outlined in Section 3, we start from approximating the validation loss up to first order around the current parameter $\theta_{t-1}$. By the first-order approximation, we shall rewrite the optimization problem in Eq. (3) as follows:

$$\max_{\substack{(w_1,\ldots,w_B),\ (\mathbf{y}_1,\ldots,\mathbf{y}_B);\\ \sum_{i=1}^{B} w_i=1,\ \mathbf{y}_i\in\Delta^K,\forall i\in[B]}} \left\langle \nabla_\theta \mathcal{L}(\mathcal{D}_{\mathrm{val}},\theta_{t-1}), \nabla_\theta \mathcal{L}\big(\mathcal{D}_{\mathrm{train}}^{\mathrm{batch}} \cup \{(w_i^{\mathrm{aug}}, x_i^{\mathrm{aug}}, \mathbf{y}_i^{\mathrm{aug}})\}_{i=1}^{B}, \theta_{t-1}\big) \right\rangle \quad (4)$$

where we also explicitly write out the learnable parts in the augmented batch.

Clearly, the set of constraints, $\sum_{i=1}^{B} w_i = 1$ and $\sum_{k=1}^{K}[K\mathbf{y}_i]_k = 1$ for $\forall i \in [B]$, are linear. To show that the objective function is also linear, we consider the form of cross-entropy loss:

$$\mathcal{L}_{\mathrm{CE}}(\mathcal{D},\theta) = -\sum_{i=1}^{B} \log \frac{\exp\left[f\left(x_i\right)\right]_{y_i}}{\sum_{k=1}^{K} \exp\left[f\left(x_i\right)\right]_k} \quad (5)$$

Since $f\left(\cdot\right): \mathcal{X} \to \Delta^K$ is assumed to have the Softmax function applied on the outputs (see Section 2), we have $\sum_{k=1}^{K} \exp\left[f\left(x_i\right)\right]_k = 1$, and the cross-entropy loss can be rewritten as:

$$\mathcal{L}_{\mathrm{CE}}(\mathcal{D},\theta) = -\sum_{i=1}^{B} \log[f\left(x_i\right)]_{y_i} \quad (6)$$

When sample weights and soft labels are introduced, the cross-entropy loss becomes:

$$\mathcal{L}_{\mathrm{CE}}(\mathcal{D},\theta) = -\sum_{i=1}^{B} w_i \cdot \sum_{k=1}^{K} [\mathbf{y}_i]_k \log[f\left(x_i\right)]_k \quad (7)$$

From the above equation, we can see that the objective function $\mathcal{L}\big(\mathcal{D}_{\mathrm{train}}^{\mathrm{batch}} \cup \{(w_i^{\mathrm{aug}}, x_i^{\mathrm{aug}}, \mathbf{y}_i^{\mathrm{aug}})\}_{i=1}^{B}, \theta_{t-1}\big)$ in Eq. (4) is indeed linear with respect to sample weigths $(w_1,\ldots,w_B)$ and soft labels $(\mathbf{y}_1,\ldots,\mathbf{y}_B)$.

Given these, we conclude, the resulted optimizaiton task, Eq. (4), is a linear programming problem, which can be solved efficiently. Moreover, the set of linear constraints are independent, which means the solution for sample weight $w$ and soft labels $\mathbf{y}$ for an augmented sample $x^{\mathrm{aug}} \in \mathcal{D}_{\mathrm{aug}}^{\mathrm{batch}}$ are independent of other augmented samples and the training sample batch $\mathcal{D}_{\mathrm{train}}^{\mathrm{batch}}$. For an arbitrary augmented sample $x^{\mathrm{aug}} \in \mathcal{D}_{\mathrm{aug}}^{\mathrm{batch}}$, replacing the gradient vector on the entire batch of training and augmented samples with the gradient vector on this single augmented sample,

$$\mathcal{L}_{\mathrm{CE}}(\{(w^{\mathrm{aug}}, x^{\mathrm{aug}}, \mathbf{y}^{\mathrm{aug}})\},\theta) = w^{\mathrm{aug}} \cdot \sum_{k=1}^{K} [\mathbf{y}^{\mathrm{aug}}]_k \log[f\left(x^{\mathrm{aug}}\right)]_k \quad (8)$$

it is not hard to see that if the gradient inner product is denoted by

$$\mathbf{\Pi} = \nabla_\theta f\left(x^{\mathrm{aug}}\right)\big|_{\theta=\theta_{t-1}} \nabla_\theta \mathcal{L}(\mathcal{D}_{\mathrm{val}},\theta_{t-1}) \quad (9)$$

the optimal solution for $w^{\mathrm{aug}}$ and $\mathbf{y}^{\mathrm{aug}}$ are:

$$\mathbf{y} = \mathrm{OneHot}\left(\arg\max_k [\mathbf{\Pi}]_k\right) \quad (10)$$

where $\mathrm{OneHot}(\cdot)$ denotes one-hot encoding, and,

$$w = 1 \text{ if } \sum_{k=1}^{K} [\mathbf{\Pi}]_k \geq 0, \text{ otherwise } w = 0 \quad (11)$$

$\square$

For the adaptation of SAFLEX to contrastive losses, in Section 3, we have already shown the main idea. Here we take a closer look at some typical contrastive losses like,

$$\mathcal{L}_{\text{contrast}}(\mathcal{D}, \theta) = -\sum_{i=1}^{B} \log \frac{\exp\left(\bar{\phi}\left(x_i\right) \cdot \bar{\phi}\left(x_i^+\right) / \tau\right)}{\exp\left(\bar{\phi}\left(x_i\right) \cdot \bar{\phi}\left(x_i^+\right) / \tau\right) + \sum_{j=1}^{B-1} \exp\left(\bar{\phi}\left(x_i\right) \cdot \bar{\phi}\left(x_j^-\right) / \tau\right)} \quad (12)$$

where $x_i^+$ is the positive example of $x_i$, and $x_j^-$ is the $j$-th negative example of $x_i$. And $\bar{\phi}\left(\cdot\right)$ is the $\mathcal{L}_2$ normalized encoder, $\tau$ is the temperature. And the contrastive pre-training (which is also use for finetuning in (Goyal et al., 2023)),

$$\mathcal{L}_{\text{CLIP}}(\mathcal{D}, \theta) := \sum_{i=1}^{B} -\log \frac{\exp\left(\bar{g}\left(I_i\right) \cdot \bar{h}\left(T_i\right)\right)}{\sum_{j=1}^{B} \exp\left(\bar{g}\left(I_i\right) \cdot \bar{h}\left(T_j\right)\right)} + \sum_{i=1}^{B} -\log \frac{\exp\left(\bar{g}\left(I_i\right) \cdot \bar{h}\left(T_i\right)\right)}{\sum_{j=1}^{B} \exp\left(\bar{g}\left(I_j\right) \cdot \bar{h}\left(T_i\right)\right)}, \quad (13)$$

where $I_i$ is the image, and $T_i$ is the text for the $i$-th sample. $\bar{g}\left(\cdot\right)$ and $\bar{h}\left(\cdot\right)$ are the $\mathcal{L}_2$ normalized image and text encoders, respectively.

We confirm that one can perceive the contrastive training objectives in Eq. (12) and Eq. (13) as proxy classification tasks. Here, we posit that the batch of size $B$ can be construed as containing $B$ classes: one positive example coupled with $B-1$ negative examples. This interpretation paves the way to introduce the notion of (soft) labels over this surrogate classification task with its $B$ distinct classes.

Taking the CLIP loss as an example, we shall generalize the first term in Eq. (13) to the following:

$$\sum_{i=1}^{B} -w_i \cdot \sum_{j=1}^{B} [\mathbf{y}_i]_j \log \frac{\exp\left(\bar{g}\left(I_i\right) \cdot \bar{h}\left(T_j\right)\right)}{\sum_{k=1}^{B} \exp\left(\bar{g}\left(I_i\right) \cdot \bar{h}\left(T_j\right)\right)} \quad (14)$$

where there are $B$ proxy-classes.

Under this paradigm, the loss function remains linear concerning the soft labels and sample weights, making the methodology in Theorem 1 applicable.

## B  RELATED WORK

In this section, we compare our approach with established augmentation methods, including traditional heuristical transformations, autonomous data augmentation, and methods leveraging large pretrained models or adversarial perturbation. We then discuss our methodology's connections to noise-robust learning and hyperparameter optimization. For a detailed background on our experimental tasks and other connected areas.

**Traditional data augmentation** operations are usually crafted and chosen heuristically based on domain expertise (Krizhevsky et al., 2017; Simard et al., 2003). For natural images, common transformations include random flipping, cropping, and color shifting (Shorten & Khoshgoftaar, 2019). Mixup-based (Zhang et al., 2018) augmentations like cutmix (Yun et al., 2019) enhance data diversity by merging patches from two images, which is also widely adopted for tabular datasets. Although we, like mixup, introduce soft labels, ours are not the outcome of merging two data instances. Nevertheless, traditional methods enjoy no guarantee of effectiveness or universality, limiting their applicability across varied data types and tasks.

**Autonomous data augmentation** has a rich history, while classical works generally bifurcate into AutoAugment-based and GAN-based approaches. **AutoAugment** (Cubuk et al., 2019) learns sequences of transformations to optimize classifier performance on a validation set. Subsequent works (Lim et al., 2019; Ho et al., 2019) have proposed alternative learning algorithms. Among them, (Mounsaveng et al., 2021; 2023) propose to learn the augmentation transformation using bilevel optimization at the cost that only differentiable affine transformations can be considered. Subsequently, RandAugment (Cubuk et al., 2020) demonstrates equivalent performance to AutoAugment by employing random transformation selection. However, such approaches still rely on a priori knowledge of beneficial transformations. On the other hand, **GAN-generated** images conditioned on their class can be used as augmented samples (Odena et al., 2017; Sankaranarayanan

et al., 2018; Huang et al., 2018). However, the inherent assumption of GANs, that augmented data should mimic the original distribution, often restricts potential enhancements (Shorten & Khoshgoftaar, 2019).

**Pretrained large generative models**, like **diffusion models**, offer the capability to synthesize training data in zero or few shot scenarios (He et al., 2022; Shipard et al., 2023) as well as generate hard training examples (Jain et al., 2022). Nonetheless, models exclusively trained on diffusion-produced data often underperform compared to their counterparts trained on real datasets (Azizi et al., 2023). To address this, recent studies (Dunlap et al., 2023; Trabucco et al., 2023) proposed the use of diffusion models for image editing with text prompts, yielding augmentations closer to original training data without necessitating finetuning. In contrast to conventional GAN-based methods, **diffusion-based augmentations** leverage knowledge from large pretrained datasets. However, they can sometimes produce subtle image edits and corrupt class-relevant information, highlighting the importance of noise reduction techniques like filtering (Dunlap et al., 2023). While such filtering relies on heuristic metrics, it can be viewed as a specific case of learning sample weights in our work. Another line of research models augmentation as **adversarial perturbations** (Goodfellow et al., 2015), aiming to generate more challenging positive and negative samples (Yang et al., 2022a;b; Ho & Nvasconcelos, 2020). However, these models usually suffer from inherent complexity and instability issues.

**Noise robust learning** bears relevance to our approach since we treat upstream augmented samples as noisy data. Learning sample weights and soft labels parallel noise reduction strategies such as dataset resampling (Han et al., 2018; Lang et al., 2022), loss reweighting (Thulasidasan et al., 2019; Konstantinov & Lampert, 2019; Gao et al., 2022), and label correction (Ma et al., 2018; Kremer et al., 2018). Our method is efficient yet principled as we formulate to optimize the model performance on the validation set, similar to standard **hyperparameter search** paradigms. Our algorithm bears relevance to continuous hyperparameter optimization Lorraine et al. (2020) in its use of bilevel optimization algorithms (Liu et al., 2022), but we introduce a novel bilevel approach. Data augmentation is greedily learned in our formulation, in sync with the ongoing training dynamics.

**Medical image classification** MedMNIST (Yang et al., 2023; 2021) is a comprehensive dataset of biomedical images, offering both 2D and 3D standardized images pre-processed to small sizes with classification labels. ResNets (He et al., 2016) popular models for medical image classification. (Yang et al., 2022a) and (Mounsaveng et al., 2023) are augmentation methods that have been shown to improve performance on medical image classification tasks. (Yang et al., 2022a) introduces a novel, prior-free autonomous data augmentation approach that leverages representation learning to create hard positive examples as augmentations, enhancing performance in various machine learning tasks without the need for a separate generative model. (Mounsaveng et al., 2023) proposes an automatic data augmentation learning method for histopathological images, wherein the augmentation parameters are determined as learnable using a bilevel optimization approach, proving more efficient and effective than predefined transformations. However, (Mounsaveng et al., 2023) is not evaluated on MedMNIST and the adaptation is non-trivial.

**Tabular data classification** Classical models, such as XGBoost (Chen & Guestrin, 2016), have been the cornerstone for tabular data processing, providing interpretability and handling diverse feature types effectively, including those with missing values. Multilayer perceptrons (MLPs) have also been a staple in the domain, offering flexibility in modeling non-linear relationships in tabular datasets. TabNet (Arik & Pfister, 2021), a more recent innovation, employs neural networks to emulate decision trees, focusing selectively on specific features at every layer. Lastly, SAINT (Somepalli et al., 2022) presents a hybrid deep learning solution tailored for tabular data. It employs attention mechanisms over both rows and columns and introduces an improved embedding technique. On tabular data, cutmix (Yun et al., 2019) is widely adopted and considered as a standard augmentation method.

**Diffusion-model-based image augmentations** Recent studies have shed light on the prowess of diffusion models in image augmentations. (Dunlap et al., 2022) introduces ALIA, a technique integrating both vision and language models. Using natural language descriptions of a dataset's classes or domains, ALIA edit the image using image-to-image diffusion models (Brooks et al., 2023), ensuring the augmented data is not only visually consistent with the original but also encompasses a broader range of diversity, particularly beneficial for fine-grained classification tasks. (Trabucco et al., 2023) propose to change the inherent semantics of images, generalizing to novel visual con-

cepts from a few labeled examples, making it especially valuable for tasks demanding semantic diversification.

**Robust finetuning of vision models**, particularly the cutting-edge variants, has witnessed significant progress in recent times. Notably, image-text pre-trained models like CLIP (Radford et al., 2021) have heralded unprecedented levels of robustness, as demonstrated in CLIP and subsequent studies (Wortsman et al., 2022; Kumar et al., 2021). While standard fine-tuning methodologies possess substantial potential, there's evidence to suggest that they might diminish robustness, especially in zero-shot paradigms. The culmination of methodologies from (Kumar et al., 2021) (LP-FT) and (Wortsman et al., 2022) (weight ensembling) represents a notable benchmark in the literature. Meanwhile, the approach by (Goyal et al., 2023) introduces a nuanced strategy to the fine-tuning landscape. Harnessing a simple yet effective technique that mimics contrastive pretraining, it casts downstream class labels as text prompts and then optimizes the contrastive loss between image embeddings and these prompt embeddings, terming it "contrastive finetuning". This method has achieved remarkable results, outstripping benchmarks in multiple areas such as distribution shifts, transfer learning, and few-shot learning. Especially on the WILDS-iWILDCam, the FLYP approach championed by (Goyal et al., 2023) has set new performance standards, surpassing both traditional finetuning and existing state-of-the-art approaches. The research solidifies contrastive finetuning as a premier, intuitive strategy for the supervised finetuning of image-text models like CLIP.

**Supervised learning via contrastive loss** has taken center stage in recent research undertakings. The methodology advocates for the fine-tuning of zero-shot models in a fashion similar to their pre-training phase by capitalizing on contrastive loss. Various studies, such as (Khosla et al., 2020) have investigated this concept in a fully supervised setting without the support of a pre-trained model. In contrast, (Gunel et al., 2020) ventured into the realm of fine-tuning vast language models, while (Zhang et al., 2021) concentrated on vision-only models. A salient distinction in the approach becomes evident when considering the addition of loss functions: while certain works have paired contrastive loss with cross-entropy, it has been observed that integrating cross-entropy with FLYP loss might negatively impact results. Direct comparisons between the two loss functions have showcased the superior accuracy credentials of contrastive loss over cross-entropy.

**Generalization aspects and theoretical understanding of data augmentation** is a less explored area. Data augmentation plays a pivotal role in boosting performance, especially in tasks such as image and text classification. (Wu et al., 2020) delves into the reasons behind the efficacy of various augmentations, specifically linear transformations, within the context of over-parametrized linear regression. The study reveals that certain transformations can either enhance estimation by expanding the span of the training data or act as regularization agents. Based on these insights, the authors present an augmentation strategy that tailors transformations to the model's uncertainty about the transformed data, validating its potency across image and text datasets. On the other hand, (Lin et al., 2022) offers a fresh perspective on data augmentation (DA), challenging traditional beliefs. While classic augmentations, like translations in computer vision, are thought to create new data from the same distribution, this fails to explain the success of newer techniques that dramatically shift this distribution. The study introduces a theoretical framework that posits that DA imposes implicit spectral regularization, achieved through manipulating the eigenvalues of the data covariance matrix and boosting its entire spectrum via ridge regression. This framework provides a profound understanding of DA's varying impacts on generalization, serving as a foundational platform for innovative augmentation design.

**Other augmentation methods that use soft labels and sample weights.** There is a recent paper, Soft-Augmentation (Liu et al., 2023), which also considers soft labels/targets and soft sample weights (i.e., loss reweighting). However, we believe there are huge methodological differences between the two methods in how they model the soft labels and weights. These methodological distinctions lead to significant differences in applicability. Below, we elaborate on the methodological and applicability differences between the two approaches and provide empirical comparisons to further highlight the novelty and improved performance of our method.

Our SAFLEX employs a learnable, augmentation-method agnostic, and more automatic and principled approach for generating soft labels and sample weights. In Soft-Augmentation, the authors implement a specific approach to generating soft labels, namely through label smoothing. Label smoothing modifies the indicator value "1" (representing the ground-truth class label) with $p = 1 - \alpha(\phi)$, where the adaptive smoothing factor $\alpha(\phi)$ is determined by the degree/strength $\phi$

of the specific sampled augmentation applied to input $x_i$. Notably, the remaining probability mass $\alpha(\phi)$ is uniformly distributed across all other class labels. The formula of $\alpha(\cdot)$ requires human modeling with domain expertise. And since different upstream augmentation methods have different definitions of the strength factor $\phi$, remodeling of $\alpha(\cdot)$ for each new augmentation method is required. The discussion in Soft-Augmentation mainly focuses on crop augmentations on images, which impressively draws insights from human visual classification experiments. Our SAFLEX, in contrast, differs in these key aspects: (a) Flexible Soft Labels: SAFLEX employs a more flexible approach to modeling soft labels, moving beyond label smoothing's limitations. We believe that uniformly distributing the probability mass across all classes may not always be the most effective strategy. This limitation of Soft-Augmentation is also acknowledged in the paper. (b) Learned Soft Labels and Sample Weights: In SAFLEX, both soft labels and sample weights are learned from a bilevel optimization problem, which is agnostic to the type and strength of the upstream augmentation method. (c) Bilevel Optimization Problem: SAFLEX confronts the inherent challenge of soft augmentation by framing it as a bilevel optimization problem. This approach represents the first rigorous formulation of the problem, underscoring an important theoretical contribution. Additionally, we introduce novel and efficient algorithms specifically designed to tackle this bilevel optimization challenge.

Our SAFLEX approach offers broader applicability compared to the Soft-Augmentation method. Unlike Soft-Augmentation, which requires an explicit augmentation strength parameter $\phi$, SAFLEX seamlessly integrates with any upstream data augmentation mechanism, including diffusion models that lack the strength parameter $\phi$. This versatility enables SAFLEX to effectively handle a wider range of data types, including medical and tabular data. SAFLEX demonstrates its versatility by effectively handling a variety of tasks, including (standard) classification, fine-grained classification, out-of-distribution (OOD) generalization, and self-supervised learning. This broad applicability is evident in our comprehensive experiments. Conversely, Soft-Augmentation primarily focuses on image classification, with specific emphasis on model occlusion performance and calibration error, thus limiting its applicability to a narrower range of tasks.

## C  EXPERIMENTAL SETUPS AND IMPLEMENTATION DETAILS

In this section, we provide more details about the experimental setups and implementation details.

The experiments are conducted on 4 NVIDIA Tesla V100 GPUs with 32GB memory each.

For the hyperparameter setting of SAFLEX algorithm, we usually set the penalty coefficient $\beta = 0$, and only set it $\beta = 1$ for experiments on the tabular datasets. We often keep the temperature $\tau = 0.01$, and only set it to be $\tau = 0.1$ on the CLIP finetuning experiment. We do not conduct hyperparameter search for the hyperparameters of SAFLEX algorithm, and we believe the performance can be further improved by hyperparameter search.

We then describe the infomation of datasets. The information about tabular datasets are listed below.

| Dataset | Task | # Features | # Categorical | # Continuous | Dataset Size | # Positives | # of Neg. | % of Positives |
|---|---|---|---|---|---|---|---|---|
| Appetency | Binary | 39 | 3 | 36 | 494,021 | 97,278 | 396,743 | 19.69 |
| Arrhythmia | Binary | 226 | 0 | 226 | 452 | 66 | 386 | 14.60 |
| Click | Binary | 12 | 7 | 5 | 39,948 | 6,728 | 33,220 | 16.84 |
| Credit | Binary | 29 | 0 | 29 | 284,807 | 492 | 284,315 | 0.17 |
| QSAR | Binary | 41 | 0 | 41 | 1,055 | 356 | 699 | 33.74 |
| Shrutime | Binary | 11 | 3 | 8 | 10,000 | 2,037 | 7,963 | 20.37 |
| Volkert | Multiclass (10) | 147 | 0 | 147 | 58,310 | — | — | — |

Table 5: Statistics of the tabular datasets.

The specific subsets of iWILDCam and CUB datasets used in diffusion-generated augmentation experiments are adopted form (Dunlap et al., 2023).

Next, we show some more experiment results. The performance of Soft-Augmentation (Liu et al., 2023) on MedMNIST datasets is listed below. Since the Soft-Augmentation paper focuses on improving crop augmentation and does not provide formulas to generate soft labels and sample weights

| Dataset | Download Link |
|---------|---------------|
| Appetency | http://kdd.ics.uci.edu/databases/kddcup99 |
| Arrhythmia | http://odds.cs.stonybrook.edu/arrhythmia-dataset/ |
| Click | https://kdd.org/kdd-cup/view/kdd-cup-2012-track-2 |
| Credit | https://www.kaggle.com/jacklizhi/creditcard |
| QSAR | https://archive.ics.uci.edu/ml/datasets/QSAR+biodegradation |
| Shrutime | https://www.kaggle.com/shrutimechlearn/churn-modelling |
| Volkert | http://automl.chalearn.org/data |

Table 6: Links of the tabular datasets.

for the upstream augmentations we considered, we test it with crop augmentation on the MedM-NIST medical image datasets. We use the tuned hyperparameters for crop augmentation and Soft-Augmentation as described in the paper.

| Method | Path | Derma | Tissue | Blood | OCT | OrganA | OrganC | OrganS |
|--------|------|-------|--------|-------|-----|--------|--------|--------|
| Crop | $92.68 \pm 0.82$ | $76.61 \pm 0.14$ | $67.38 \pm 0.19$ | $95.38 \pm 0.12$ | $77.50 \pm 0.11$ | $94.46 \pm 0.14$ | $90.29 \pm 0.09$ | $80.19 \pm 0.06$ |
| Soft Augmentation (w/ Crop) | $91.95 \pm 0.59$ | $77.05 \pm 0.24$ | $67.06 \pm 0.44$ | $95.96 \pm 0.28$ | $76.92 \pm 0.46$ | $93.90 \pm 0.25$ | $91.44 \pm 0.24$ | $80.92 \pm 0.17$ |

Table 7: Soft-Augment's performance on MedMNIST images, ResNet-18 backbone is used.

We see that except on the DermaMNIST dataset, the performance of Soft-Augmentation is even lower than the No Augmentation baseline. While SAFLEX's performance is consistently higher than the 'No Augmentation' baseline. Applying crop augmentation directly decreases the performance on most of the MedMNIST datasets. This is not surprising as we observed that applying RandAugment or Mixup directly also lowers the performance. However, the main reason for Soft-Augmentation's relatively poor performance is that it cannot consistently improve performance over the crop augmentation baseline (it shows improvement on Derma, Blood, OrganC, OrganS, but decreases performance on Path, Tissue, OCT, OrganA). This suggests that in situations with a high prevalence of poor-quality augmented samples (e.g., crop augmentation on medical images), Soft-Augmentation's relatively conservative strategy is inadequate in overcoming the significant noise and label errors introduced by these samples.

The performance of LP-A3 (Yang et al., 2022a) on MedMNIST datasets (copied from the original paper) is listed below for reference.

| Method | Path | Derma | Tissue | Blood | OCT | OrganA | OrganC | OrganS |
|--------|------|-------|--------|-------|-----|--------|--------|--------|
| LP-A3 | $94.42 \pm 0.24$ | $76.22 \pm 0.27$ | $68.63 \pm 0.14$ | $96.97 \pm 0.06$ | $80.27 \pm 0.54$ | $94.73 \pm 0.21$ | $92.41 \pm 0.22$ | $82.28 \pm 0.38$ |

Table 8: LP-A3's performance on MedMNIST images, ResNet-18 backbone is used.

For the efficiency result, we found that on the eight MedMNIST datasets considered, the overhead of SAFLEX measure as wall-clock time is 42% on average, while more specifically, 54% with RandAugment and 31% with Mixup. On the seven tabular datasets, on average the overhead of SAFLEX is 81% of the original training time per epoch.