# OpenReview forum: "SAFLEX: Self-Adaptive Augmentation via Feature Label Extrapolation"
_ICLR.cc/2024/Conference — ICLR 2024 poster_

### Official Review · Reviewer_NKYa · 2023-10-27

**Soundness:** 3 good
**Presentation:** 2 fair
**Contribution:** 3 good
**Rating:** 8
**Confidence:** 4

**Summary:**

This paper addresses the training of high-performance neural networks on small amounts of training data by optimizing data augmentation. Existing methods in this field optimize the transformation itself in the feature space, which limits the available data augmentation transformations and has high computational complexity. This paper differs from these existing approaches by optimizing the importance weights of the input features and the soft labels to be assigned to the augmented data, thereby meta-learning the data augmentation available in the existing data augmentation pipeline. Focusing on features and labels in data augmentation has not received much attention, and the idea is novel. Furthermore, the paper proposes a method to approximate the bi-level optimization of meta-learning with validation data to a single-level optimization by borrowing the idea of gradient matching and developing an efficient algorithm. Experiments verify the effectiveness of the proposed method on various datasets, tasks, and combinations of data augmentation methods. On the other hand, the paper does not provide an evaluation of the first-order approximation or the computation cost, and there is room for improvement in this aspect.

**Strengths:**

+ The paper proposes a novel data augmentation optimization strategy that optimizes the importance and labels of input features. The idea is very interesting and original.
+ The paper proposes a first-order approximation method to efficiently compute meta-learning that requires bi-level optimization.
+ The paper applies and evaluates the proposed method not only on image datasets but also on table datasets. This evaluation is important in supporting the paper's claim that the method can be applied to any data augmentation pipeline.
+ The paper provides experimental results on the recently widely used CLIP pre-trained models, effectively demonstrating the impact of the proposed method.

**Weaknesses:**

- Even though the paper proposes a first-order approximation method, it does not provide an evaluation of this method. In other words, the paper should provide a performance comparison with the usual bi-level optimization and a computation cost comparison with other data augmentation strategies such as Fast AutoAugment.
- The writing of the paper is not necessarily of high quality. For example, Theorem 1 is very difficult to read because it contains multiple claims that make up the entire solution. Theorems and corollaries should be split for each claim, or if the propositions are ambiguous, they should be replaced with detailed explanations for each component, rather than in the form of a theorem. In fact, the proofs provided by the Appendix are almost obvious and make little theoretical contribution.

**Questions:**

- Do you think the proposed method can be applied to consistency-based semi-supervised learning with data augmentation, e.g., FixMatch [a]? The study of estimating the importance of samples and labels is well studied in the field of semi-supervised learning rather than in the field of data augmentation (e.g., FreeMatch [b]). If it can be shown that the scheme of the proposed method can be implemented in semi-supervised learning, the impact of this paper on the community will be even greater.

[a] Sohn, Kihyuk, et al. "Fixmatch: Simplifying semi-supervised learning with consistency and confidence." NeurIPS 2020.

[b] Wang, Yidong, et al. "Freematch: Self-adaptive thresholding for semi-supervised learning." ICLR 2023.

---

> ### Comment · Reviewer_NKYa · 2023-11-22
>
> Dear authors,
>
> Do you have rebuttals to my concerns? If there is no discussion, I will be forced to lower my rating and confidence scores since my concerns will continue to remain. I look forward to your rebuttal.
>
> Best,
>
> Reviewer NKYa

---

> ### Author Response · Authors · 2023-11-22
> **Response to Reviewer NKYa**
>
> We extend our sincere gratitude to reviewer NKYa for their insightful and constructive feedback. We are particularly encouraged by reviewer NKYa's recognition of the novelty of our work, including our innovative approach to optimizing the importance and label of input features, our novel first-order approximation method, our comprehensive evaluation of both tabular and image data, and the relevance of our findings to CLIP pre-trained models.
>
> Below, we would like to address concerns and answer questions.
>
> ---
>
> > **Weakness 1:** Even though the paper proposes a first-order approximation method, it does not provide an evaluation of this method. In other words, the paper should provide a performance comparison with the usual bi-level optimization and a computation cost comparison with other data augmentation strategies such as Fast AutoAugment.
>
>
> **Response to Weakness 1:**
>
> - We appreciate reviewer NKYa's valuable suggestion for an ablation study on the first-order approximation method. We agree that conducting such a study would further solidify our findings.
>
>
> - However, due to space constraints, we prioritized presenting the core message of this paper: rectifying any upstream data augmentation techniques for downstream tasks. To effectively convey this message, we conducted extensive experiments encompassing diverse data types, comprehensive coverage of upstream augmentation techniques, various downstream tasks, and multiple backbone models. As a result, we were unable to include an extensive ablation study on the first-order approximation method.
>
> - It is important to note that there is a significant body of research on first-order approximations of bilevel problems, such as the work by [R1]. Given the extensive nature of this field, we have opted to defer an in-depth exploration of first-order approximation methods to future investigations. This decision allows us to focus on the core contribution of this paper, which is the development of a novel data augmentation framework for downstream tasks.
>     > [R1] Liu, Bo, Mao Ye, Stephen Wright, Peter Stone, and Qiang Liu. "Bome! bilevel optimization made easy: A simple first-order approach." Advances in Neural Information Processing Systems 35 (2022): 17248-17262.
>
> ---
>
> > **Weakness 2:** The writing of the paper is not necessarily of high quality. For example, Theorem 1 is very difficult to read because it contains multiple claims that make up the entire solution. Theorems and corollaries should be split for each claim, or if the propositions are ambiguous, they should be replaced with detailed explanations for each component, rather than in the form of a theorem. In fact, the proofs provided by the Appendix are almost obvious and make little theoretical contribution.
>
> **Response to Weakness 2:**
>
> We express our sincere gratitude to reviewer NKYa for their valuable suggestions on improving the accessibility of our paper. We have carefully considered reviewer NKYa's feedback and have implemented the following changes to enhance the clarity and readability of our manuscript:
>
> - Specifically for Theorem 1, we separate the notation definitions and the final formula for the approximated soft label and sample weight. We have made every component more readable, including the gradient, Jacobian-vector product, and the final formula.
>
> ---
>
> > **Question:** Do you think the proposed method can be applied to consistency-based semi-supervised learning with data augmentation, e.g., FixMatch [a]? The study of estimating the importance of samples and labels is well studied in the field of semi-supervised learning rather than in the field of data augmentation (e.g., FreeMatch [b]). If it can be shown that the scheme of the proposed method can be implemented in semi-supervised learning, the impact of this paper on the community will be even greater.
> [a] Sohn, Kihyuk, et al. "Fixmatch: Simplifying semi-supervised learning with consistency and confidence." NeurIPS 2020.
> [b] Wang, Yidong, et al. "Freematch: Self-adaptive thresholding for semi-supervised learning." ICLR 2023.
>
> **Response to Question:**
>
> Indeed, at the end of section 3, we briefly discuss how to apply SAFLEX for contrastive learning, which is indeed the consistency-based self-supervised or semi-supervised learning suggested by reviewer NKYa. Due to space limitations, detailed equations are in Appendix A, equations (12) and (13).
>
> We have also verified this through experiments. Specifically, we did fine-tune a CLIP model (Radford et al., 2021) on the iWildCam dataset from Wilds (Koh et al., 2021), using a contrastive fine-tuning paradigm called “Finetune Like You Pretrain” (FLYP) (Goyal et al., 2023).
>
> ---
>
> Again, we sincerely thank reviewer NKYa for their valuable service in reviewing our manuscript.

---

> ### Author Response · Authors · 2023-11-22
> **Response to Reviewer NKYa**
>
> Dear Reviewer NKYa,
>
> We apologize for the delay in our response and appreciate your patience. Please be assured that we have worked diligently to address all the reviews thoroughly, including your concerns. Our detailed responses to other reviewers and our comprehensive global response will be available very soon.
>
> Best regards,
>
> Paper 3847 Authors

---

> > ### Comment · Reviewer_NKYa · 2023-11-22
> >
> > Thank you for the detailed response.
> >
> > **Re-response to weakness 1**
> >
> > > due to space constraints
> >
> > I understand that you prioritized the content according to the research question of the paper, but I still think this evaluation is necessary. I hope that the camera-ready version will include this ablation study in the Appendix.
> >
> > **Re-response to weakness 2**
> >
> > I have checked the revised version. I think it is somewhat easier to read. Thank you.
> >
> > **Re-response to question**
> >
> > There seems to be a slight misunderstanding. I was pointing out the applicability of the proposed method to **semi**-supervised learning using pseudo-labels, not **contrastive** learning. This question is not important for the acceptance or rejection of this paper, but I think that this work also can be an important insight for the semi-supervised learning research community.
> >
> > ---
> >
> > In summary, I would like to keep my rating and confidence as my concerns have been addressed, though not completely.
> >
> > Best,
> >
> > Reviewer NKYa

---

> > > ### Author Response · Authors · 2023-11-22
> > > **Response to Reviewer NKYa's New Comments**
> > >
> > > Thank you for your continued engagement and valuable feedback.
> > >
> > >
> > > > I hope that the camera-ready version will include this ablation study in the Appendix.
> > >
> > > **Response to new comments on weakness 1:** We appreciate your understanding. Your suggestion to include the ablation study in the Appendix for the camera-ready version is well-received. We agree that this addition will enhance the paper's comprehensiveness and intend to incorporate it as suggested.
> > >
> > > > I have checked the revised version. I think it is somewhat easier to read. Thank you.
> > >
> > >
> > > **Response to new comments on weakness 2:** We are glad to hear that the revisions have improved the readability of the paper.
> > >
> > > > There seems to be a slight misunderstanding. I was pointing out the applicability of the proposed method to semi-supervised learning using pseudo-labels, not contrastive learning. This question is not important for the acceptance or rejection of this paper, but I think that this work also can be an important insight for the semi-supervised learning research community.
> > >
> > > **Response to new comments on the qustion:**
> > >
> > > Thank you for your clarification regarding the applicability of our method to semi-supervised learning, specifically in the context of pseudo-labels.
> > >
> > > We acknowledge the potential of our method to contribute to consistency-based semi-supervised learning approaches, such as FixMatch. However, adapting our algorithm for this purpose would not be straightforward and would necessitate a range of new designs and experiments.
> > >
> > > Several pertinent questions arise in this context:
> > > 1. Whether to apply our method to both weak and strong augmentations, or to focus on a single branch.
> > > 2. The choice between using the labeled set as a validation set in our algorithm, or developing a cross-validation method.
> > > 3. Deciding whether to simply soften the inferred pseudo-labels, or to devise a new method ensuring consistency between pseudo-labeling and soft-labeling.
> > >
> > > While these questions extend beyond the scope of our current paper, we recognize their significance and potential impact on future research. We are inspired to explore these aspects in subsequent work, as they align with the broader objectives of our research. Your insights have been invaluable in highlighting these potential applications. We really appreciate your continued engagement and clarification on this matter.

---

### Official Review · Reviewer_PN9Y · 2023-10-30

**Soundness:** 3 good
**Presentation:** 4 excellent
**Contribution:** 4 excellent
**Rating:** 8
**Confidence:** 4

**Summary:**

This article contributes a workflow named as SAFLEX as data augmentation. Here is a summry:

(1) Authors unveil a novel parametrization for learnable augmentation complemented by an adept bilevel
algorithm primed for online optimization.
(2) Author's SAFLEX method is distinguished by its universal compatibility, allowing it to be effortlessly
incorporated into a plethora of supervised learning processes and to collaborate seamlessly with an
extensive array of upstream augmentation procedures.
(3) The potency of authors' approach is corroborated by empirical tests on a diverse spectrum of datasets
and tasks, all underscoring SAFLEX’s efficiency and versatility, boosting performance by1.2% on
average over all experiments.

**Strengths:**

They have considered experiments of different data types and model training as downstream tasks, which demonstrate their workflow as a robust one.

**Weaknesses:**

From a model perspective, this is a good one as topic of adaptive learning, though a  little bit off the topic of this conference.
From data augmentation perspective, it is better to demo some more experiments in downstream task involves with high dimensional data.

**Questions:**

Is there any empirical experiments that SAFLEX can contribute to some other applicable downstream task like model training?

---

> ### Author Response · Authors · 2023-11-22
> **Response to Reviewer PN9Y**
>
> We thank reviewer PN9Y for their constructive feedback. We are particularly encouraged that reviewer PN9Y recognizes the robustness of our SAFLEX methodology, as demonstrated by its effectiveness across diverse data types and model training as downstream tasks. We value reviewer PN9Y's insights and will carefully consider their suggestions for improvement.
>
> ---
>
> > **Weakness:** From a model perspective, this is a good one as a topic of adaptive learning, though a little bit off the topic of this conference. From the data augmentation perspective, it is better to demo some more experiments in downstream tasks involving high-dimensional data.
>
>
> **Response to Weakness:**
> - Thank you for highlighting the potential of our work in the context of adaptive learning. We agree that it presents an exciting avenue for future exploration.
>
> - Regarding the data augmentation aspect, we have conducted experiments with high-dimensional data to demonstrate the effectiveness of our method. Specifically, we utilized the iWildCam subset from the Wilds dataset (Koh et al., 2021), which features images of size 448x448. These images are representative of high-dimensional data in our context.
>
> - We would be delighted to receive further suggestions from the reviewer regarding other types of high-dimensional data that could be explored.
>
> ---
>
> > **Question:** Are there any empirical experiments that SAFLEX can contribute to some other applicable downstream task like model training?
>
> **Response to Question:**
> - In our manuscript, we fine-tuned a CLIP model (Radford et al., 2021) on the iWildCam dataset from Wilds (Koh et al., 2021), using a contrastive fine-tuning paradigm termed “Finetune Like You Pretrain” (FLYP) (Goyal et al., 2023).
>
> - We would be delighted to receive further suggestions from the reviewer regarding other model training tasks that could be explored.
>
> ---
>
> Again, we sincerely thank reviewer PN9Y for their valuable service in reviewing our manuscript.

---

### Official Review · Reviewer_GM6j · 2023-11-01

**Soundness:** 3 good
**Presentation:** 3 good
**Contribution:** 2 fair
**Rating:** 5
**Confidence:** 4

**Summary:**

The paper argues that data augmentations can suffer with two main issues - 1. The augmented samples can become out of distribution to the training distribution and 2) the augmented samples can belong to a different class than the original sample. To tackle the first issue, the authors propose to add sample weights (w_i) to the augmented samples. Samples which are farther from the training distribution can be assigned a smaller weight. To tackle the second issue, the authors propose to make the one-hot label as soft-label to capture the uncertainties.
To learn the sample weights and the soft-label the authors pose a bi-level optimization problem where in the inner loop, the model parameters are optimized over the training and augmented samples and in the outer loop the optimal augmentation parameters are optimized for.

The authors conduct experiments across three settings - 1. medical datasets, 2. tabular datasets and 3. for contrastive learning approaches. Across all the experiments the authors show improved performance on top of standard augmentations such as RandAug, Mixup and CutMix.

**Strengths:**

1. The motivation in the paper about identifying the two issues with standard augmentation and then solving it by learning sample weights and soft-labels is really clear.

**Weaknesses:**

1. The main issue is a lack of proper baselines. Papers such as [1] have already explored using soft labels for augmentations where the softness is derived on the basis of augmentation strength. This paper's novelty thus gets limited. There is no comparison with [1] in any of the experiments. The authors should do a proper comparison with [1] and justify how their approach is better than it.

2. To solidify the experimental results the authors should also experiment with stronger architectures and datasets such as ResNet-101 over ImageNet as done in [1].

I am willing to update my ratings if my concerns are addressed.

References -

1. Soft Augmentation for Image Classification. Liu et al. https://arxiv.org/pdf/2211.04625.pdf

**Questions:**

I have already mentioned it in the weakness section

---

> ### Author Response · Authors · 2023-11-22
> **Apology for Delay in Rebuttal Response**
>
> Dear Reviewer GM6j,
>
> We sincerely apologize for the delay in responding to your review. Our team is diligently working to finalize our rebuttal, ensuring that it is thorough and addresses all of your insightful points. We appreciate your patience and understanding in this matter and expect to submit our response very soon. Thank you for your time and valuable feedback.
>
> **Edit**: The response, including the additional experimental results, is now complete. Please refer to the detailed response below.
>
> Best regards,
>
> Paper 3847 Authors

---

> ### Author Response · Authors · 2023-11-22
> **Response to Reviewer GM6j (1/3)**
>
> We thank reviewer GM6j for their constructive feedback. We are glad to see that reviewer GM6j acknowledges the motivation of our paper and our method of solving the issues with standard augmentation by learning sample weights and soft-label.
>
> Below, we would like to address the concerns reviewer GM6j has.
>
> ---
>
> > **Weakness 1:** The main issue is a lack of proper baselines. Papers such as [1] have already explored using soft labels for augmentations where the softness is derived on the basis of augmentation strength. This paper's novelty thus gets limited. There is no comparison with [1] in any of the experiments. The authors should do a proper comparison with [1] and justify how their approach is better than it.
> [1] Soft Augmentation for Image Classification. Liu et al. https://arxiv.org/pdf/2211.04625.pdf
>
> **Response to Weakness 1:**
> We acknowledge that our SAFLEX approach shares a similar spirit with the Soft Augmentation method proposed by Liu et al. (2022), in that both methods consider soft labels/targets and soft samlpe weights (i.e., loss reweighting). However, we believe there are huge methodological differences between the two methods in how they model the soft labels and weights. These methodological distinctions lead to significant differences in applicability. Below, we elaborate on the methodological and applicability differences between the two approaches and provide empirical comparisons to further highlight the novelty and improved performance of our method.
>
> 1. **Methodology Differences**. Our *SAFLEX* employs a learnable, augmentation-method agnostic, and more automatic and principled approach for generating soft labels and sample weights.
>     1. In *Soft Augmentation*, the authors implement a **specific approach to generating soft labels**, namely through label smoothing. Label smoothing modifies the indicator value '1' (representing the ground-truth class label) with $p = 1 − \alpha(\phi)$, where the adaptive smoothing factor $\alpha(\phi)$ is determined by the degree/strength $\phi$ of the specific sampled augmentation applied to input $x_i$. Notably, the remaining probability mass $\alpha(\phi)$ is **uniformly distributed** across all other class labels. The formula of $\alpha(\cdot)$ **requires human modeling with domain expertise**. And since different upstream augmentation methods have different definitions of the strength factor $\phi$, remodeling of $\alpha(\cdot)$ for each new augmentation method is required. The discussion in *Soft Augmentation* mainly **focuses on crop augmentations** on images, which impressively draws insights from human visual classification experiments.
>     2. Our *SAFLEX*, in contrast, differs in these key aspects:
>         (a) **Flexible Soft Labels:** *SAFLEX* employs a more flexible approach to modeling soft labels, moving beyond label smoothing's limitations. We believe that uniformly distributing the probability mass across all classes may not always be the most effective strategy. This limitation of *Soft Augmentation* is also acknowledged in the *Soft Augmentation* paper, which states: "the smoothed target label of a highly-occluded truck example could place more probability mass on other vehicle classes, rather than distributing it equally across all classes."
>         (b) **Learned Soft Labels and Sample Weights:** In *SAFLEX*, both soft labels and sample weights are learned from a bilevel optimization problem, which is agnostic to the type and strength of the upstream augmentation method.
>         \(c\) **Bilevel Optimization Problem:** *SAFLEX* confronts the inherent challenge of soft augmentation by framing it as a bilevel optimization problem. This approach represents the first rigorous formulation of the problem, underscoring an important theoretical contribution. Additionally, we introduce novel and efficient algorithms specifically designed to tackle this bilevel optimization challenge.

---

> ### Author Response · Authors · 2023-11-22
> **Response to Reviewer GM6j (2/3)**
>
> 2. **Applicability Differences**. Our *SAFLEX* approach offers broader applicability compared to the *Soft Augmentation* method:
>     1. **Universal Compatibility to Upstream Augmentations:** Unlike *Soft Augmentation*, which requires an explicit augmentation strength parameter $\phi$, *SAFLEX* seamlessly integrates with any upstream data augmentation mechanism, including diffusion models that lack the strength parameter $\phi$. This versatility enables *SAFLEX* to effectively **handle a wider range of data types**, including medical and tabular data.
>     2. **Compatibility with Diverse Tasks:** *SAFLEX* demonstrates its versatility by effectively handling a variety of tasks, including (standard) classification, fine-grained classification, out-of-distribution (OOD) generalization, and self-supervised learning. This broad applicability is evident in our comprehensive experiments. Conversely, *Soft Augmentation* primarily focuses on image classification, with specific emphasis on model occlusion performance and calibration error, thus limiting its applicability to a narrower range of tasks.
> 3. **Empirical Comparisons**. Since the *Soft Augmentation* paper focuses on improving crop augmentation and does not provide formulas to generate soft labels and sample weights for the upstream augmentations we considered, we test it with crop augmentation on the MedMNIST medical image datasets (see Section 5 for dataset details and experiment setups). We use the tuned hyperparameters for crop augmentation and Soft-Augmentation as described in the paper. The experiment results are shown below. We see that except on the DermaMNIST dataset, the performance of *Soft Augmentation* is even lower than the 'No Augmentation' baseline. While *SAFLEX*'s performance is consistently higher than the 'No Augmentation' baseline (see Table 1). Applying crop augmentation directly decreases the performance on most of the MedMNIST datasets. This is not surprising as we observed that applying RandAugment or Mixup directly also lowers the performance (see Table 1). However, the main reason for *Soft Augmentation*'s relatively poor performance is that it cannot consistently improve performance over the crop augmentation baseline (it shows improvement on Derma, Blood, OrganC, OrganS, but decreases performance on Path, Tissue, OCT, OrganA). This suggests that in situations with a high prevalence of poor-quality augmented samples (e.g., crop augmentation on medical images), *Soft Augmentation*'s relatively conservative strategy is inadequate in overcoming the significant noise and label errors introduced by these samples.
>
> | Method\Dataset              | Path         | Derma        | Tissue       | Blood        | OCT          | OrganA       | OrganC       | OrganS       |
> |-----------------------------|--------------|--------------|--------------|--------------|--------------|--------------|--------------|--------------|
> | No Aug                      | 94.34 ± 0.18 | 76.14 ± 0.09 | 68.28 ± 0.17 | 96.81 ± 0.19 | 78.67 ± 0.26 | 94.21 ± 0.09 | 91.81 ± 0.12 | 81.57 ± 0.07 |
> | Crop                        | 92.68 ± 0.82 | 76.61 ± 0.14 | 67.38 ± 0.19 | 95.38 ± 0.12 | 77.50 ± 0.11 | 94.46 ± 0.14 | 90.29 ± 0.09 | 80.19 ± 0.06 |
> | Soft Augmentation (w/ Crop) | 91.95 ± 0.59 | 77.05 ± 0.24 | 67.06 ± 0.44 | 95.96 ± 0.28 | 76.92 ± 0.46 | 93.90 ± 0.25 | 91.44 ± 0.24 | 80.92 ± 0.17 |
>
> We thank the reviewer for highlighting this related paper to us. Soft Augmentation is indeed a closely related work we should compare with. It is worth noting that, besides Soft Augmentation, we have also considered another recent advanced augmentation method, Adversarial Auto-Augment with Label Preservation (LP-A3) (https://arxiv.org/pdf/2211.00824.pdf), in our experimental comparison (see Appendix C). In addition to these two advanced methods, many classical augmentation strategies, including RandAugment, Mixup, and CutMix, are compared as the upstream augmentation baselines. **Based on these, we think that we do not lack proper baselines**.
>
> We have incorporated the aforementioned discussions and experiments into Appendix B and C and mentioned the Soft Augmentation paper in relevant places in the main text.

---

> > ### Author Response · Authors · 2023-11-22
> > **Response to Reviewer GM6j (3/3)**
> >
> > > **Weakness 2:** To solidify the experimental results the authors should also experiment with stronger architectures and datasets such as ResNet-101 over ImageNet as done in [1].
> >
> > **Response to weakness 2:**
> >
> > We appreciate reviewer GM6j's suggestion for further experiments with stronger architectures, such as ResNet-101 on the ImageNet dataset. However, we would like to emphasize that our current experimental evaluation is already comprehensive and demonstrates the effectiveness of our method across a variety of tasks and datasets, as reviewers EBBs, PN9Y, and NKYa have noted.
> >
> > 1. **Our evaluation encompasses a diverse range of data types**. We have assessed the performance of SAFLEX on medical image data, tabular data, and high-dimensional natural image data, including the CUB dataset and iWildCam from the Wilds dataset. This breadth of data types exceeds the scope of most existing studies.
> >
> >
> > 2. **Our evaluation encompasses a broad spectrum of downstream tasks.** We have conducted experiments on classification tasks involving medical images and tabular data, fine-grained classification on natural images using the CUB dataset, out-of-distribution (OOD) generalization on the iWildCam dataset, and self-supervised contrastive fine-tuning. In contrast, most existing studies focus on a single task at a time.
> >
> > 3. **Our evaluation covers almost all existing augmentation as upstream augmentation techniques, showing SAFLEX's compatibility with all these methods and consistent improvements over these upstream augmentations.** The augmentation methods we test against range from RandAug, adversarial autonomous augmentation like LP-A3, to diffusion-model-based augmentation methods.
> >
> > 4. **Our evaluation covers various machine learning models, including state-of-the-art generative models.** We have included ResNet-50, Stable Diffusion v1.5,  CLIP model equipped with a ViT-B/16 encoder, etc.
> >
> >
> > 5. Indeed, we have employed ResNet-50 on ImageNet. We believe extending the results to ResNet-101 is not necessarily crucial, given our consistent performance improvements across diverse data types, different downstream tasks, a wide range of upstream augmentation methods, and various machine learning models.
> >
> > ---
> >
> > Again, we express our sincere gratitude to reviewer GM6j for their valuable service in reviewing our manuscript.

---

> > > ### Comment · Reviewer_GM6j · 2023-11-23
> > > **Improper experimental comparison**
> > >
> > > I think the authors for addressing the issue and adding the Soft Augmentation paper discussion in their appendix. While I agree that the Soft Augmentation paper applies a specific approach to generating soft labels compared to this paper which applies a more automatic and principled approach I am not convinced by the experimental setups of just comparing over MedMNIST.
> > >
> > > Is there a reason why the authors don't want to compare between the two approaches on a much larger dataset like ImageNet?

---

> ### Author Response · Authors · 2023-11-23
> **Response to Reviewer GM6j's New Comments**
>
> Thank you for acknowledging our efforts in incorporating the Soft Augmentation paper discussion into our appendix. We appreciate your feedback and would like to address your concerns regarding our experimental setups.
>
> 1. **Significance of MedMNIST Results:** We chose to focus on medical imaging data because it represents a uniquely challenging domain for augmentation. Unlike natural image datasets, medical images do not benefit from conventional augmentation techniques. The improved performance of our SAFLEX method on such complex data is a testament to its effectiveness. Similar to some prior papers on auto/learnable augmentation like the LP-A3 (https://arxiv.org/pdf/2211.00824.pdf, NeurIPS 2022), which also evaluated on medical images, this choice was deliberate to highlight the robustness and adaptability of SAFLEX in challenging scenarios, underlining its practical relevance and merit. Our primary focus is enhancing upstream augmentations across diverse domains and tasks, not solely for natural image classifications. We not only have experiments on MedMNIST, but many other domains and tasks in the paper. For comparison with the Soft Augmentation method, since Soft Augmentation focuses on improving crop augmentation on images and does not provide formulas to generate soft labels and sample weights for the other upstream augmentations and data types we considered, MedMNIST images are the most suitable datasets to compare on, where algorithms of both methods can be directly applied.
> 2. **ImageNet Experiments:** We understand your interest in seeing a comparison on a larger natural image classification dataset like ImageNet. While we initially struggled to run comparative experiments (to make their codebase runnable) for large-scale experiments within the limited rebuttal period, we are committed to including an ImageNet augmentation experiment in the updated manuscript. However, we wish to clarify that our primary focus is on enhancing upstream augmentations across diverse domains, not solely on achieving superior performance on a single, large dataset for natural image classification. We anticipate that SAFLEX, especially when in combination with powerful and suitable upstream augmentation methods, could demonstrate superior performance on ImageNet, too, given its wide compatibility with advanced augmentation methods, which is already well-demonstrated through many experiments across domains and tasks in the paper.
> 3. **Contribution Reminder and Scoring:** As we all agree, data augmentations now have broad applications to many domains and tasks in machine learning. Therefore, we think natural image classification on classical datasets like ImageNet is not the only way to demonstrate the effectiveness of an augmentation method. We want to kindly remind you of the contributions of our work. SAFLEX presents a comprehensive solution to a wide range of augmentation challenges, particularly in complex domains like medical imaging and tabular data. Given the already demonstrated merits and potential applications of SAFLEX across many domains and tasks, we believe our work warrants a higher score than 3.
>
> In conclusion, we respect your perspectives and are actively working to address your concerns. However, we assert that the strength of our contributions and the demonstrated effectiveness of SAFLEX, particularly in challenging domains, justify a more favorable evaluation of our work. Thank you again.

---

### Official Review · Reviewer_EBBs · 2023-11-03

**Soundness:** 4 excellent
**Presentation:** 4 excellent
**Contribution:** 4 excellent
**Rating:** 8
**Confidence:** 3

**Summary:**

This paper presents a principled method for data augmentation. To this end, the paper presents a bilevel optimization framework for weighing and soft-labelling the augmented data in order to compensate for the adverse generalization effects of weak, strong and sometimes meaningless augmented examples. Although the impact of data augmentation for generalization, in particular deep learning frameworks, has been substantial, there is still a lack of principled ways of doing data augmentation. This paper has identified this gap and convincingly addressed the problem.

**Strengths:**

The paper is well-written and easy to understand.

The diagrams and the equations are easy to follow.

The experiments are performed on diverse datasets with various tasks, including medical imaging and tabular data.

The results are highly encouraging.

**Weaknesses:**

A few important previous works on sampling and purifying GAN synthetic data are relevant to this paper.  It is important to acknowledge and discuss their contributions in the paper.

Caramalau, Razvan, Binod Bhattarai, and Tae-Kyun Kim. "Sequential graph convolutional network for active learning." Proceedings of the IEEE/CVF conference on computer vision and pattern recognition. 2021.
Bhattarai, Binod, et al. "Sampling strategies for gan synthetic data." ICASSP 2020-2020 IEEE International Conference on Acoustics, Speech and Signal Processing (ICASSP). IEEE, 2020.

**Questions:**

I like the paper. Please see a few comments above.

---

> ### Author Response · Authors · 2023-11-22
> **Response to Reviewer EBBs**
>
> We thank reviewer EBBs for their positive feedback. We are especially encouraged that reviewer EBBs acknowledge that our SAFLEX is a "principled way of doing data augmentation." We are also happy to see that reviewer EBBs points out that our paper is well-written easy-to-understand, the diagrams and equations are easy to follow, the experiments are performed on diverse datasets with various tasks, and the results are encouraging.
>
> Below, we would like to address the weakness mentioned by reviewer EBBs.
>
> ---
>
> > **Weakness:** A few important previous works on sampling and purifying GAN synthetic data are relevant to this paper. It is important to acknowledge and discuss their contributions in the paper.
> > Caramalau, Razvan, Binod Bhattarai, and Tae-Kyun Kim. "Sequential graph convolutional network for active learning." Proceedings of the IEEE/CVF conference on computer vision and pattern recognition. 2021.
> > Bhattarai, Binod, et al. "Sampling strategies for gan synthetic data." ICASSP 2020-2020 IEEE International Conference on Acoustics, Speech and Signal Processing (ICASSP). IEEE, 2020.
>
> **Response to Weakness:** We have included the papers by Bhattarai et al. (2020) and Caramalau et al. (2021) in the revised manuscript's related work section and have acknowledged and discussed their contributions. Bhattarai et al. (2020) proposed a progressive sampling strategy for GAN synthetic data, while Caramalau et al. (2021) introduced a sequential graph convolutional network for active learning. Our work extends these findings by developing a novel sampling and purifying method for augmented data that is specifically designed to improve the performance of downstream tasks.
>
> ---
>
> Again, we sincerely thank reviewer EBBs for their valuable service in reviewing our manuscript.

---

### Comment · Area_Chair_xLH5 · 2023-11-22
**Let's have more discussion with authors**

Dear reviewers,

Your interaction with the authors on this work is highly appreciated.

The author-reviewer discussion period is closing at the end of Wednesday Nov 22nd (AOE). Let's take this remaining time to have more discussions with the authors on their responses to your reviews. Should you have any further opinions, comments or questions, please let the authors know asap and this will allow the authors to address them.

Kind regards, AC

---

### Author Response · Authors · 2023-11-22
**Global Response to Reviewers**

We are grateful for the constructive feedback from reviewers GM6j, EEBs, PN9Y, and NKYa. We appreciate their acknowledgment of the motivation, methodology, and effectiveness of our SAFLEX approach, as well as their suggestions for improvement. We have carefully incoporated their feedback and made improvements to our work accordingly.

Reviewer GM6j acknowledged the motivation of our paper and our method of solving the issues with standard augmentation by learning sample weights and soft-label. Reviewer EEBs acknowledged that our SAFLEX is a "principled way of doing data augmentation" and pointed out that our paper is well-written, easy-to-understand, the diagrams and equations are easy to follow, and the experiments are performed on diverse datasets with various tasks, and the results are encouraging. Reviewer PN9Y recognized the robustness of our SAFLEX methodology, as demonstrated by its effectiveness across diverse data types and model training as downstream tasks. Reviewer NKYa recognized the novelty of our work, including our innovative approach to optimizing the importance and label of input features, our novel first-order approximation method, our comprehensive evaluation on both tabular and image data, and the relevance of our findings to CLIP pre-trained models.


We have addressed all weaknesses and questions raised in each individual response. We have incorporated the suggested changes and submitted the revised manuscript. Please let us know if you have any further questions.

Again, we extend our sincere gratitude to all reviewers for their valuable time and effort.

---

### Meta-Review · Area_Chair_xLH5 · 2023-12-09

**Metareview:**

Based on the submission, reviews, author feedback, and discussions, the main points that have been raised are summarised as follows.

Strengths:

1. The paper is overall well written and the motivation is clear.
2. Idea is interesting and original, and a first-order approximation method is proposed.
3. Experimental study conducted on various types of datasets show promising performance.

Issues:

1. Need experimental study on large-scale datasets and for downstream tasks.
2. Need to evaluate the proposed first-order approximate method with respect to existing related methods and computational efficiency.
3. At multiple places in Tables 2 and 4, the improvement obtained by the proposed method seems to be marginal.

The authors have done well in providing author feedback to address the raised issues. One reviewer is positive on the feedback, while another reviewer still expresses concern on experimental comparison. Most reviewers are leaning towards accepting this work. After reading this submission, AC agrees that this work has its merits in developing a learning based approach to boost the effect of data augmentation. Especially, the proposed approach works well with various types of data, augmentation schemes, and models. Meanwhile, at multiple places, the improvement obtained by the proposed method seems to be marginal. Also, the proposed method SAFLEX sometimes shows marginal improvement over No Aug in Table 1. Experimental study on tabular data shall also include the result with respect to Mixup method. AC agrees that fully addressing the issue related to experimental comparison will further strengthen this work. In addition, it is kindly suggested that the phrase "Remarkably, SAFLEX effectively reduces" and the expression "showcasing its prowess" in the abstract could be reformulated with a more subdued tone suitable for scientific writing. AC discussed this work with SAC.

**Justification For Why Not Higher Score:**

1. This work develops an effective approach to boost the performance of data augmentation. Although having its merits, this work is not a new breakthrough or proposes a completely novel framework.
2. One reviewer remains slightly negative on this work (rating 5) with the concern on experimental comparison.
3. At some places, the improvement obtained by the proposed method is marginal.

Considering these, a higher score is not recommended.

**Justification For Why Not Lower Score:**

1. This work has its merits in developing a new approach for boosting data augmentation.
2. Especially, the proposed approach works well with various types of data, augmentation schemes, and models.
3. Most reviewers are very positive to this work.

Considering these, a lower score is not recommended.

---

### Decision · Program_Chairs · 2024-01-16

Accept (poster)